# Cloud fraction determined by thermal infrared and visible all-sky cameras

Christine Aebi[1,2], Julian Gröbner[1], and Niklaus Kämpfer[2]

[1]Physikalisch-Meteorologisches Observatorium Davos, World Radiation Center, Davos, Switzerland.
[2]Oeschger Center for Climate Change Research and Institute of Applied Physics, University of Bern, Bern, Switzerland.

*Correspondence to:* Aebi Christine (christine.aebi@pmodwrc.ch)

**Abstract.** The thermal infrared cloud camera (IRCCAM) is a prototype instrument that determines cloud fraction continuously during day and nighttime using measurements of the absolute thermal sky radiance distributions in the 8 $\mu$m - 14 $\mu$m wavelength range in conjunction with clear sky radiative transfer modelling. Over a time period of two years, the fractional cloud coverage obtained by the IRCCAM is compared with two commercial cameras (Mobotix Q24M and Schreder VIS-J1006) sensitive in the visible spectrum, as well as with the automated partial cloud amount detection algorithm (APCADA) using pyrgeometer data. Over the two year period, the cloud fractions determined by the IRCCAM and the visible sky cameras are consistent to within 2 oktas (0.25 cloud fraction) for 90 % of the dataset during the day while for day- and nighttime data the comparison with the APCADA algorithm yields an agreement of 80 %. These results are independent of cloud types with the exception of thin cirrus clouds which are not detected as consistently by the current cloud algorithm of the IRCCAM. The measured absolute sky radiance distributions also provide the potential for future applications by combining these measurements with ancillary meteorological data from radiosondes and ceilometers.

## 1 Introduction

Clouds affect the surface radiation budget and thus the climate system on a local as well as on a global scale. Clouds have an influence on solar and on terrestrial radiation by absorbing, scattering and emitting radiation. The Intergovernmental Panel on Climate Change (IPCC) states that clouds in general and aerosol-cloud interactions in particular generate considerable uncertainty in climate predictions and climate models (IPCC, 2013). Having information about cloud fraction on a local scale is of importance in different fields: for solar power production due to the fact that clouds cause a large variability in the energy production (Parida et al., 2011; Mateos et al., 2014; Tzoumanikas et al., 2016), for aviation and weather forecast or microclimatological studies.

The most common practice worldwide to determine cloud coverage, cloud base height (CBH) and cloud type from the ground are human observations (CIMO, 2014). These long-term series of cloud data allow climate studies to be conducted (e.g. Chernokulsky et al., 2017). Cloud detection by human observers is carried out several times per day over a long time period without the risk of a larger data gap due to a technical failure of an instrument. However, even with a reference standard defined by the World Meteorological Organisation (WMO) for human observers, the cloud determination is not objective e.g. mainly

due to varying degrees of experience (Boers et al., 2010). Other disadvantages of human cloud observations are that the temporal resolution is coarse and due to visibility issues nighttime determinations are difficult. Since clouds are highly variable in space and time, measurements at high spatial and temporal resolution with small uncertainties are needed (WMO, 2012). Recent research has therefore been conducted to find an automated cloud detection instrument (or a combination of such) to

replace human observers (Boers et al., 2010; Tapakis and Charalambides, 2013; Huertas-Tato et al., 2017; Smith et al., 2017). An alternative to detect clouds from the ground by human observations is to detect them from space. With a temporal resolution of 5 to 15 minutes, Meteosat Second Generation (MSG) geostationary satellites are able to detect cloud coverage with a higher time resolution than is accomplished by human observers (Ricciardelli et al., 2010; Werkmeister et al., 2015). The geostationary satellite Himawari-8 (Da, 2015) even delivers cloud information with a temporal resolution of 2.5 to 10 min-

utes and a spacial resolution of 0.5 to 2 km. However, these geostationary satellites cover only a certain region of the globe. Circumpolar satellites (i.e. the MODIS satellites Terra and Aqua (Baum B.A., 2006; Ackerman et al., 2008)) determine cloud fraction globally, but for a specific region only four times a day. Satellites cover a larger area than ground-based instruments and are also able to deliver cloud information from regions where few ground-based instruments are available (e.g. in Arctic regions (Heymsfield et al., 2017) or over oceans). However, due to the large field of view (FOV) of satellites, small clouds

can be overlooked (Ricciardelli et al., 2010). Another challenge with satellite data is the ability to distinguish thin clouds from land (Dybbroe et al., 2005; Ackerman et al., 2008). Furthermore, satellites collect information mainly from the highest cloud layer rather than the lower cloud layer closer to the earth's surface. Nowadays satellite data are validated and thus supported by ground-based cloud data. Different studies focusing on the comparison of the determined cloud fraction from ground and from space were presented by e.g. Fontana et al. (2013); Wacker et al. (2015); Calbo et al. (2016); Kotarba (2017).

In general, three automatic ground-based cloud cover measurement techniques are distinguished: radiometers, active column instruments and hemispherical sky cameras. Radiometers measure the incident radiation in different wavelength ranges. Depending on the wavelength range, the presence of clouds alters the radiation measured at ground level (e.g. Calbo et al., 2001; Mateos Villàn et al., 2010). Calbo et al. (2001) and Dürr and Philipona (2004) both present different methodologies to determine cloud conditions from broadband radiometers. Other groups describe methodologies using instruments with a smaller

spectral range. Such instruments are for example the infrared pyrometer CIR-7 (Nephelo) (Tapakis and Charalambides, 2013) or Nubiscope (Boers et al., 2010; Feister et al., 2010; Brede et al., 2017), which both measure in the 8 $\mu$m - 14 $\mu$m wavelength range of the spectrum. In order to retrieve cloud information, Nephelo consists of seven radiometers which scan the whole upper hemisphere. The Nubiscope consists of one radiometer only, which also scans the whole upper hemisphere. Such a scan takes several minutes, which is a limitation on the retrieval of cloud fraction information when for example fast-moving clouds

occur (Berger et al., 2005). In general, these instruments give information about cloud fraction for three different levels, cloud types and cloud base height (CBH) (Wauben, 2006). Brocard et al. (2011) presents a method using data from the tropospheric water vapour radiometer (TROWARA) to determine cirrus clouds from the measured fluctuations in the sky infrared brightness temperature.

The second group, the column cloud detection instruments send a laser pulse to the atmosphere and measure the backscattered

photons. The photons are scattered back by hydrometeors in clouds and, depending on the time and the amount of backscat-

tered photons measured, the cloud base height can be determined. However, the laser pulse is not only scattered back by cloud hydrometeors, but also by aerosols (Liu et al., 2015). Examples of active remote sensing instruments are cloud radar (Kato et al., 2001; Illingworth et al., 2007; Feister et al., 2010), lidar (Campbell et al., 2002; Zhao et al., 2014) and ceilometers (Martucci et al., 2010). Due to the narrow beam, a disadvantage of these measurement techniques is the lack of instantaneous

cloud information of the whole upper hemisphere. Boers et al. (2010) showed that with smaller integration times the instruments tend to give okta values of zero and eight rather than the intermediate cloud fractions of 1 to 7 oktas.

The third group of ground-based cloud detection instruments comprises the hemispherical sky cameras, which have a 180° view of the upper hemisphere. The most common all-sky camera is the commercially available total sky imager (TSI) (Long et al., 2006). Another pioneering hemispherical cloud detection instrument is the whole sky imager (WSI) (Shields et al., 2013).

Whereas the TSI is sensitive in the visible spectrum, the WSI acquires information in seven different spectral ranges in the visible and in the near infrared regions. A special version of the WSI also allows nighttime measurements (Feister and Shields, 2005). Other cloud research has been undertaken with low-cost commercial cameras sensitive in the visible spectrum of the wavelength range (e.g. Calbo and Sabburg, 2008; Cazorla et al., 2008; Kazantzidis et al., 2012; Wacker et al., 2015; Kuhn et al., 2017). All these hemispherical sky cameras operate well during daytime, but give no information during nighttime. Thus, there

is increasing interest in development of cloud cameras sensitive in the thermal infrared region of the spectrum. Ground-based thermal infrared all-sky cameras have the advantage of delivering continuous information about cloud coverage, cloud base height and cloud type during day and nighttime, which in turn is of interest in various fields.

The infrared cloud imager (ICI) is a ground-based sky camera sensitive in the 8 $\mu$m - 14 $\mu$m wavelength range and with a resolution of 320 $\times$ 240 pixels (Shaw et al., 2005; Thurairajah and Shaw, 2005; Smith and Toumi, 2008). Another instrument,

the Solmirus all-sky infrared visible analyser (ASIVA) consists of two cameras, one measuring in the visible and the other one in the 8 $\mu$m - 13 $\mu$m wavelength range (Klebe et al., 2014). The whole-sky infrared cloud measuring system (WSIRCMS) is an all-sky cloud camera sensitive in the 8 $\mu$m - 14 $\mu$m wavelength range (Liu et al., 2013). The WSIRCMS consists of nine cameras measuring at the zenith and at eight surrounding positions. With a time resolution of 15 minutes, information about cloud cover, CBH and cloud type are determined. This instrument has an accuracy of $\pm 0.3$ oktas compared to visual observations

(Liu et al., 2013). Redman et al. (2018) presented a reflective all-sky imaging system (sensitive in the 8 $\mu$m - 14 $\mu$m wavelength range) consisting of a longwave infrared microbolometer camera and a reflective sphere (110° FOV). The Sky Insight thermal infrared cloud imager is an industrial and patented (Bertin et al., 2015b) product from Reuniwatt. The Sky Insight cloud imager is sensitive in the 8 $\mu$m - 13 $\mu$m wavelength range and gives cloud information of the whole upper hemisphere. Their system is mainly used for cloud cover forecasts up to 30 minutes ahead, which is relevant for e.g. global horizontal irradiance forecasts

or optical communication link availability (Bertin et al., 2015a; Liandrat et al., 2017).

The current study describes a newly developed prototype instrument, the thermal infrared cloud camera (IRCCAM), that consists of a modified commercial thermal camera (Gobi-640-GigE) that gives instantaneous information about cloud conditions for the full upper hemisphere. The time resolution of the IRCCAM in the current study is 1 minute during day- and nighttime. It measures in the wavelength range of 8 $\mu$m - 14 $\mu$m. After a developing and testing phase (Aebi et al., 2014; Gröbner et al.,

2015), the IRCCAM is in continuous use at the Physikalisch-Meteorologisches Observatorium Davos/World Radiation Center

(PMOD/WRC), Davos, Switzerland, since September 2015. The IRCCAM was developed to provide instantaneous hemispheric cloud coverage information from the ground with a high temporal resolution in a more objective way than human cloud observations. Thus the IRCCAM could be used for different applications at meteorological stations, at airports or at solar power plants. The performance of the IRCCAM regarding cloud fraction is compared with data from two visible all-sky

cameras and the automatic partial cloud amount detection algorithm (APCADA) (Dürr and Philipona, 2004). In section 2, the instruments and cloud detection algorithms are presented. The comparison of the calculated cloud fractions based on different instruments and algorithms are analysed and discussed overall and for different cloud classes, times of day and seasons separately in section 3. Section 4 provides a summary and conclusions.

## 2    Data and Methods

All three all-sky camera systems used for the current study are installed at the Physikalisch-Meteorologisches Observatorium Davos/World Radiation Center (PMOD/WRC), Davos, located in the Swiss Alps (46.81°N, 9.84°E, 1,594 m asl). There are two commercial cameras, one Q24M from Mobotix and the other is a VIS-J1006 cloud camera from the company Schreder. Both of these cameras are measuring in the visible spectrum. The third camera is the newly developed all-sky camera (IRCCAM) sensitive in the thermal infrared wavelength range. The instruments themselves and their respective analysis software are

described in the following subsections. Also, the automatic partial cloud amount detection algorithm (APCADA) is briefly described in Section 2.4.

The analysis of the data from the thermal infrared cloud camera (IRCCAM) is performed for the time period September 21, 2015 to September 30, 2017, with a data gap between December 20, 2016 and February 24, 2017 due to maintenance of the instrument. Mobotix and APCADA data are available for the whole aforementioned time period. Schreder data are only

available since March 9, 2016. Thus the analysis of these data is only performed for the time period March 9, 2016 to September 30, 2017.

### 2.1    Thermal infrared cloud camera

The Infrared Cloud Camera (IRCCAM) (Figure 1) consists of a commercial thermal infrared camera (Gobi-640-GigE) from Xenics (www.xenics.com). The camera is an uncooled microbolometer sensitive in the wavelength range of 8 $\mu$m - 14 $\mu$m. The

chosen focal length of the camera objective is 25 mm and the field of view $18° \times 24°$. The image resolution is $640 \times 480$ pixels. The camera is located on top of a frame looking downward on a gold-plated spherically shaped aluminium mirror such that the entire upper hemisphere is imaged on the camera sensor. The complete system is 1.9 m tall. The distance between the camera objective and the mirror is about 1.2 m. These dimensions were chosen in order to reflect the radiation from the whole upper hemisphere onto the mirror and to minimise the area of the sky hidden by the camera itself. The arm holding the

camera above the mirror is additionally fixed with two wire ropes to stabilise the camera during windy conditions. The mirror is gold-plated to reduce the emissivity of the mirror and to make measurements of the infrared sky radiation largely insensitive to the mirror temperature. Several temperature probes are included to monitor the mirror, camera and ambient temperatures.

The camera of the IRCCAM was calibrated in the PMOD/WRC laboratory in order to determine the brightness temperature or the absolute radiance in $Wm^{-2}sr^{-1}$ for every pixel in an IRCCAM image. The absolute calibration was obtained by placing the camera in front of the aperture of a well characterised blackbody at a range of known temperatures between -20 °C and +20 °C in steps of 5 °C (Gröbner, 2008). The radiance emitted by a blackbody radiator can be calculated using the Planck radiation

formula,

$$L_\lambda(T) = \frac{2hc^2}{\lambda^5} \frac{1}{e^{\frac{hc}{k\lambda T}} - 1} \tag{1}$$

where $T$ is the temperature, $\lambda$ the wavelength, $h$ is the Planck constant, $6.6261 \times 10^{-34}$ Js, $c$ the speed of light, 299'792'458 ms$^{-1}$ and $k$ the Boltzmann constant, $1.3806 \times 10^{-23}$ J K$^{-1}$. For the IRCCAM camera, the spectral response function $R_\lambda$ as provided by the manufacturer is shown in Figure 2 and is used to calculate the integrated radiance $L_R$,

$$L_R = \int_8^{25} R_\lambda \cdot L_\lambda(T) d\lambda \tag{2}$$

where $T$ is the effective temperature of the blackbody (Gröbner, 2008) and $L_R$ the integrated radiance measured by the IRC-CAM camera. To retrieve the brightness temperature ($T_B$) from the integrated radiance $L_R$, Eq. 2 cannot be solved analytically. Therefore, as an approximation, we are using a polynomial function $T_B = f(L_R)$ to retrieve the brightness temperature $T_B$ from the radiance $L_R$. Using Eq. 2, $L_R$ values are calculated for temperatures in the range of -40 °C and +40 °C. The resulting

fitting function is a polynomial third order function (see Figure 3), which is used to retrieve $T_B$ from the integrated radiance $L_R$ for every pixel in an IRCCAM image.

The IRCCAM calibration in the blackbody aperture was performed on March 16, 2016 and all its images are calibrated with the corresponding calibration function retrieved from the laboratory measurements. The calibration uncertainty of the camera in terms of brightness temperatures (in a range of -40 °C and +40 °C) is estimated at 1 K for a Planck spectrum as emitted

by a blackbody radiator. Furthermore, a temperature correction function for the camera was derived from these laboratory calibrations in order to correct the measurements obtained at ambient temperatures outdoors.

The hemispherical sky images taken by the IRCCAM are converted to polar coordinates ($\Theta$, $\Phi$) for the purpose of retrieving brightness temperatures in dependence of zenith and azimuth respectively. Due to slight aberrations in the optical system of the IRCCAM, the $\Theta$ coordinate does not follow a linear relationship with the sky zenith angle, producing a distorted sky image.

Therefore, a correction function was determined by correlating the apparent solar position as measured by the IRCCAM with the true solar position obtained by a solar position algorithm. This correction function was then applied to the raw camera images to obtain undistorted images of the sky hemisphere.

One should note that observing the sun with the Gobi camera implies that the spectral filter used in the camera to limit the spectral sensitivity to the 8 $\mu$m - 14 $\mu$m wavelength band has some leakage at shorter wavelengths. Fortunately, this leakage is

confined to a narrow region around the solar disk (around 1°) as shown in Figure 4. Thus it has no effect on the remaining part of the sky images taken by the IRCCAM during daytime measurements.

The main objective of the IRCCAM study is to determine cloud properties from the measured sky radiance distributions. The

cloudy pixels in every image are determined from their observed higher radiances with respect to that of a cloud-free sky. The clear sky radiance distributions are determined from radiative transfer calculations using MODTRAN 5.1 (Berk et al., 2005), using as input parameters screen-level air temperature and integrated water vapour (IWV). The temperature was determined at 2 m elevation obtained from a nearby SwissMetNet station, while the IWV was retrieved from GPS signals operated by the Federal Office for Topography and archived in the Studies in Atmospheric Radiation Transfer and Water Vapour Effects (STARTWAVE) database hosted at the Institute of Applied Physics at the University of Bern (Morland et al., 2006). For practical reasons, a lookup table (LUT) for a range of temperatures and IWV was generated which was then used to compute the reference clear sky radiance distribution for every single image taken by the camera. A similar approach to detect cloud patterns is described in Bertin et al. (2015a) and Liandrat et al. (2017).

The sky brightness temperature distribution as measured on a cloud-free day (June 18, 2017 10:49 UTC) and the corresponding modelled sky brightness temperature are shown in Figure 4a and Figure 4b, respectively. As expected, the lowest radiance is emitted at the zenith, with a gradual increase at increasing zenith angle, until the measured effective sky brightness temperature at the horizon is nearly equal to ambient air temperature (Smith and Toumi, 2008). Figure 4c shows the profiles of the measured (red) and modelled (blue) brightness temperatures along one azimuth position going through the solar position (yellow line in Figure 4a). As can be seen in Figure 4c, the measured and modelled sky distributions agree fairly well, with large deviations at high zenith angles due to the mountains obstructing the horizon around Davos. The shortwave leakage from the sun can also be clearly seen around pixel number 180. A smaller deviation is seen at pixel number 239 from the wires holding the frame of the camera.

The average difference between the measured and modelled clear sky radiance distributions was determined for several clear sky days during the measurement period in order to use that information when retrieving clouds from the IRCCAM images. Such differences can arise on the one hand from the rather crude radiative transfer modelling which only uses surface temperature and IWV as input parameters to the model. On the other hand it can arise from instrumental effects such as a calibration uncertainty of $\pm1$ K. An effect of the mirror temperature and a possible mismatch between actual and nominal spectral response functions of the IRCCAM camera are other potential causes for this difference. But both of these possible effects have not been taken into account. The validation measurements span 8 days, with full sky measurements obtained every minute, yielding a total of 11,512 images for the analysis. For every image, the corresponding sky radiance distribution was calculated from the LUT, as shown in Figure 4b. The residuals between the measured and modelled sky radiance distributions were calculated by averaging over all data points with zenith angles smaller than $60°$, while removing the elements (frame and wires) of the IRCCAM within the field of view of the camera, resulting in one value per image. The brightness temperature differences between IRCCAM and model calculations show a mean difference of +4.0 K and a standard deviation of 2.4 K over the whole time period. The observed variability comes equally from day-to-day variations as well as from variations within a single day. No systematic differences are observed between day and nighttime data.

The stability of the camera over the measurement period is investigated by comparing the horizon brightness temperature derived from the IRCCAM with the ambient air temperature measured at the nearby SwissMetNet station. As mentioned by Smith and Toumi (2008), the horizon brightness temperature derived from the IRCCAM should approach the surface air

temperature close to the horizon. Indeed, the average difference between the horizon brightness temperature derived from the IRCCAM and the surface air temperature was 0.1 K with a standard deviation of 2.4 K, showing no drifts over the measurement period and thus confirming the good stability of the IRCCAM during this period. The good agreement of 0.1 K between the derived horizon brightness temperature from the IRCCAM and the surface air temperature confirms the absolute calibration uncertainty of ± 1 K of the IRCCAM. Therefore, the observed discrepancy of 4 K between measurements and model calculations mentioned previously can probably be attributed to the uncertainties in the model parameters (temperature and IWV) used to produce the LUT.

### 2.1.1 Cloud detection algorithm

After setting up the IRCCAM, a horizon mask is created initially to determine the area of the IRCCAM image representing the sky hemisphere. A cloud-free image is selected manually. The sky area is selected by the very low sky brightness temperatures with respect to the local obstructions with much larger brightness temperatures. This image mask contains local obstructions such as the IRCCAM frame (camera, arm and wire ropes) as well as the horizon, which in the case of Davos consists of mountains limiting the field of view of the IRCCAM. Thereafter, the same horizon mask is applied to all IRCCAM images. The total number of pixels within the mask is used as a reference and the cloud fraction is defined as the number of pixels detected as cloudy relative to the total number.

The algorithm to determine cloudy pixels from an IRCCAM image consists of two parts. The first part uses the clear sky model calculations as a reference to retrieve low to mid-level clouds. These clouds have large temperature differences compared to the clear sky reference. In this part of the algorithm, cloudy pixels are defined for measured sky brightness temperatures that are at least 6.5 K greater than the modelled clear-sky reference value. A rather large threshold value was empirically chosen to avoid any erroneous clear sky mis-classifications as cloudy pixels. The thinner and higher clouds with lower brightness temperatures are therefore left for the second part of the algorithm.

In order to determine the thin and high-level clouds within an IRCCAM image, non cloudy pixels remaining from the first part of the algorithm are used to fit an empirical clear sky brightness temperature as a function of the zenith angle,

$$T_B = (T_{65} - a) \left( \frac{\Theta}{65} \right)^b + a \tag{3}$$

where $T_B$ is the brightness temperature for a given zenith angle $\Theta$, and $T_{65}$, $a$ and $b$ are the retrieved function parameters (Smith and Toumi, 2008). This second part of the algorithm assumes a smooth variation of the clear sky brightness temperature with zenith angle. Thereby it determines cloudy pixels as deviations from this smooth function as well as requiring a brightness temperature higher than this empirical clear sky reference. Pixels with a brightness temperature higher than the empirically defined threshold of 1.2 K are defined as cloudy and removed from the clear sky data set. This procedure is repeated up to 10 times to iteratively find pixels with a brightness temperature higher than the clear sky function. One restriction of this fitting method is that it requires at least broken cloud conditions, as it does not work well under fully overcast conditions without the presence of any cloud-free pixels to constrain the fitting procedure.

The selected threshold of 1.2 K allows the detection of low emissivity clouds, but still misses the detection of thin, high-level

cirrus clouds even though they can be clearly seen in the IRCCAM images. Unfortunately, reducing the threshold to less than 1.2 K results in many clear sky mis-classifications as clouds. Therefore under these conditions, it seems that using a spatial smoothness function is not sufficient to infer individual pixels as being cloudy; a more advanced algorithm as discussed in Brocard et al. (2011) is required to define clouds not only on a pixel by pixel basis but as a continuous structure (e.g. pattern
recognition algorithm).

Before reaching the final fractional cloud data set, some data filtering procedures are applied: situations with precipitation are removed by considering precipitation measurements from the nearby SwissMetNet station; ice or snow deposition on the IRCCAM mirror is detected by comparing the median radiance of a sky area with the median radiance value of an area on the image showing the frame of the IRCCAM. In cases where the difference between the median values of the two areas is smaller
than the empirically defined value of 5 $Wm^{-2}sr^{-1}$, the mirror is assumed contaminated by snow or ice and therefore does not reflect the sky, so the image is excluded. The horizon mask does not cover all pixels that do not depict sky, which leads to an offset in the calculated cloud fraction of around 0.04. This offset is removed before comparing the cloud fraction determined by the IRCCAM with other instruments.

## 2.2   Mobotix camera

A commercial surveillance Q24M camera from Mobotix (www.mobotix.com) has been installed in Davos since 2011. The camera has a fisheye lens and is sensitive in the red-green-blue (RGB) wavelength range. The camera takes images from the whole upper hemisphere with a spatial resolution of $1200 \times 1600$ pixels. The camera system is heated, ventilated and installed on a solar tracker with a shading disk. The shading disk avoids overexposed images due to the sun. The time resolution of the
Mobotix data is one minute (from sunrise to sunset) and the exposure time is 1/500 s.

An algorithm determines the cloud fraction of each image automatically (Wacker et al., 2015; Aebi et al., 2017). Before applying the cloud detection algorithm, the images are preprocessed. The distortion of the images is removed by applying a correction function. The same horizon mask, which was defined on the basis of a cloud-free image, is applied to all images. After this preprocessing, the colour ratio (the sum of the blue to green ratio plus the blue to red ratio) is calculated per pixel.
To perform the cloud determination per pixel, this calculated colour ratio is compared to an empirically defined reference ratio value of 2.2. Comparing the calculated colour ratio value with this reference value designates whether a pixel is classified as cloudy or as cloud-free. The cloud fraction is calculated by the sum of all cloud pixels divided by the total number of sky pixels.

The cloud classes are determined with a slightly adapted algorithm from Heinle et al. (2010) which is based on statistical
features (Wacker et al., 2015, Aebi et al., 2017). The cloud classes determined are stratocumulus (Sc), cumulus (Cu), stratus-altostratus (St-As), cumulonimbus-nimbostratus (Cb-Ns), cirrocumulus-altocumulus (Cc-Ac), cirrus-cirrostratus (Ci-Cs) and cloud-free (Cf).

## 2.3 Schreder camera

The total sky camera VIS-J1006 from Schreder (www.schreder-cms.com) consists of a digital camera with a fisheye lens. The VIS-J1006 Schreder camera is sensitive in the RGB region of the spectrum and takes two images every minute with different exposure times (1/500 s and 1/1600 s, respectively). The aperture is fixed at $f$/8 for both images. The resolution of the images is $1200 \times 1600$ pixels. The camera comes equipped with a weatherproof housing and a ventilation system.

The images from the Schreder camera are analysed using two different algorithms. The original software is directly delivered from the company Schreder. Before calculating the fractional cloud coverage, some steps are needed to define the settings that are needed to preprocess the images. In a first step, the centre of the image is defined manually. In a second step, the maximum zenith angle of the area taken into account for further analyses is defined. Unfortunately, the maximum possible zenith angle is only $70°$ and thus a larger fraction of the sky cannot be analysed. After the distortion of the images is removed, in a fourth step a horizon mask is defined on the basis of a cloud-free image. The mask also excludes the pixels around the sun. In a last step, a threshold is defined which specifies whether a pixel is classified or not classified as a cloud. The settings from these preprocessing steps are then applied to all images from the Schreder camera. In the following, the term Schreder refers to data where this algorithm is used.

Due to the Schreder algorithm's limitation of a maximum zenith angle of $70°$, we used the same algorithm as for the Mobotix camera, referred hereafter as Schreder$_{pmod}$. The algorithm Schreder$_{pmod}$ has the advantage that the whole upper hemisphere is considered when calculating the fractional cloud coverage. Thus, a new horizon mask is defined on the basis of a cloud-free image. The colour ratio reference to distinguish between clouds and no clouds is assigned an empirical value of 2.5, which is slightly different to that used for the Mobotix camera. The Schreder camera in Davos has been measuring continuously since March 2016.

## 2.4 APCADA

The automated partial cloud amount detection algorithm (APCADA) determines the cloud amount in oktas using downward longwave radiation from pyrgeometers, temperature and relative humidity measured at screen-level height (Dürr and Philipona, 2004). APCADA is only able to detect low- and mid-level clouds and is not sensitive to high-level clouds. The time resolution of APCADA is 10 minutes during day and nighttime. The agreement of APCADA compared to synoptic observations at high-altitude and midlatitude stations, such as Davos, is that 82 % to 87 % of cases during day and nighttime have a maximum difference of $\pm1$ okta ($\pm0.125$ cloud fraction) and between 90 % to 95 % of cases have a difference of $\pm2$ oktas ($\pm0.250$ cloud fraction) (Dürr and Philipona, 2004).

In order to compare the cloud coverage information retrieved from APCADA with the fractional cloud coverages retrieved from the cameras, the okta values are converted to fractional cloud coverage values by multiplying the okta values by 0.125. In the current study, APCADA is mainly used for comparisons of the nighttime IRCCAM data.

## 3   Results

In the aforementioned time period September 21, 2015 to September 30, 2017, the IRCCAM data set comprises cloud cover information from 581,730 images. The Mobotix data set comprises 242,249 images (because only daytime data are available) and the Schreder data set 184,746 images (shorter time period and also only daytime). Figure 5 shows the relative frequencies

of cloud cover detection from the different camera systems in okta bins. Zero okta corresponds to a cloud fraction of 0 to 0.05 and 8 oktas to a cloud fraction of 0.95 to 1. One and seven oktas correspond to intermediate bins of 0.1375 cloud fraction and oktas two to six to intermediate bins of 0.125 cloud fraction (Wacker et al., 2015). Cloud-free (0 okta) and overcast (8 oktas) are the cloud coverages that are most often detected in the aforementioned time period. This behaviour also agrees with the analysis of the occurrence of fractional cloud coverages over a longer time period in Davos discussed in Aebi et al. (2017).

All four instruments show a similar relative occurrence of cloud coverages of 2 - 6 oktas. It is noteworthy that the IRCCAM clearly underestimates the occurrence of 0 oktas in comparison to the cameras measuring in the visible spectrum (by up to 13 %). On the other hand, the relative frequency of the IRCCAM of 1 okta is clearly larger (by up to 10 %) compared to the visible cameras. This can be explained by higher brightness temperatures measured in the vicinity of the horizon above Davos. These higher measured brightness temperatures are falsely determined as cloudy pixels (up to 0.16 cloud fraction). Since these

situations with larger brightness temperatures occur quite frequently, the IRCCAM algorithm detects more often cloud coverages of 1 okta instead of 0 okta. Also, at the other end of the scale, the IRCCAM is detecting slightly larger values of a relative frequency of 7 oktas compared to the visible cameras and slightly lower relative frequencies of a measurement of 8 oktas.

As an example, Figure 6 shows the cloud fraction determined on April 4, 2016, where various cloud types and cloud fractions were present. This day starts with an overcast sky and precipitation and therefore the IRCCAM is measuring fractional cloud

coverages of more than 0.98. The cloud layer disperses until it reaches cloud fraction values of 0.1 at around 6 UTC. At this time the sun rises above the effective horizon and the visible all-sky cameras start to measure shortly thereafter. The cloud classes are determined with the algorithm developed by Wacker et al. (2015) based on Mobotix images. In the early morning, the cloud type present is cumulus. The larger difference of more than 0.1 between the cloud fraction determined by the Schreder algorithm and the other algorithms can be explained after a visual observation of the image: the few clouds that are present are

located close to the horizon and thus in the region of the sky that the Schreder algorithm is not able to analyse. The fractional cloud coverage increases again to values of around 0.8 at 7 UTC. At this time, all four cameras and algorithms determine a similar fractional cloud coverage. Around 8 UTC a first cirrostratus-layer appears which is slightly better detected by the IRCCAM and the Mobotix algorithm than by the two algorithms using the Schreder images. Two hours later, around 10 UTC, the main cloud type present is again cumulus. Low-level clouds are quite precisely detected by all camera systems and thus, in

this situation, the maximum observed difference is only 0.06. Figure 7a shows exactly this situation as an RGB-image taken by the Mobotix camera, and the corresponding classifications as cloudy or non-cloudy pixels determined by the IRCCAM (Figure 7b) and by the Mobotix algorithm (Figure 7c). From 11 UTC onwards the cumulus clouds are found in the vicinity of the horizon and cirrus-cirrostratus closer to the zenith. Because all algorithms have difficulties to detect thin and high-level clouds, the differences in the determined cloud fractions are variable. Again, the Schreder algorithm is not able to analyse the cloud

fraction near the horizon and thus it always detects the smallest fraction compared to the other algorithms. The visible cameras continue measuring until 16:23 UTC when the sun sets and afterwards only data from the IRCCAM are available.

## 3.1 Visible all-sky cameras

Before validating the fractional cloud coverage determined by the IRCCAM algorithm, the fractional cloud coverages, which are determined using the images of the visible all-sky cameras Mobotix and Schreder, are compared among each other to gain a better understanding of their performance. The time period analysed here is March 9, 2016 to September 30, 2017, consisting of only daytime data, which corresponds to a data set of 184,746 images. Additionally, the results from the visible all-sky cameras are compared with data retrieved from APCADA (temporal resolution of 10 min). For this comparison, 32,902 and

24,907 Mobotix and Schreder images respectively are considered.

The histograms of the residuals of the difference in the cloud fractions (range between [-1;1]) between the visible all-sky cameras are shown in Figure 8 and the corresponding median and 5th and 95th percentiles are shown in Table 1.

As shown in Table 1, the two algorithms from the Schreder camera as well as APCADA underestimate the cloud fraction determined from Mobotix images, with a maximum median difference of -0.04. Although the median difference in cloud frac-

tion between the two Schreder algorithms is 0.00, the distribution tends towards more negative values. This more pronounced underestimation of fractional cloud coverage of the Schreder algorithm might be explained by the smaller fraction of the sky being analysed (Figure 8c). The underestimation in the retrieved cloud fraction of the Schreder algorithm for 90 % of the data is even slightly larger in comparison to the cloud fraction determined with the Mobotix algorithm. The spread (shown as 5th and 95th percentiles in Table 1) is greatest for all comparisons of the algorithms from the visible cameras with APCADA. As

previously mentioned in Section 2.4, APCADA gives the cloud fraction only in steps of 0.125, and is thus not as accurate as the cloud fraction determined from the cameras. This fact might explain the large variability in the residuals.

In Figure 8 it is shown that the distribution of the residuals between the cloud fraction retrieved from Mobotix versus the cloud fraction retrieved from the two Schreder algorithms (Figure 8a and 8b) are left-skewed, which confirms that the cloud fraction retrieved from the two Schreder algorithms underestimates the cloud fraction retrieved from the Mobotix images.

Taking the measurement uncertainty of human observers and also of other cloud detection instruments to be $\pm 1$ okta to $\pm 2$ oktas (Boers et al., 2010), we take this as a baseline uncertainty range to test the performance in the detection of cloud fraction of our visible camera systems. The algorithms for the visible camera systems determine the cloud fraction for 94 - 100% of the data within $\pm 2$ okta ($\pm 0.25$) and for 77 - 94 % of the data within $\pm 1$ okta ($\pm 0.125$). Comparing the cloud fraction determined from APCADA with the cloud fraction determined from the visible cameras shows that in only 62 - 71 % of the cases is there

an agreement of $\pm 1$ okta ($\pm 0.125$) and in 83 - 86 % of data an agreement of $\pm 2$ okta ($\pm 0.25$). All these results are further discussed in the next Section.

## 3.2 IRCCAM Validation

As described in Section 3.1, in up to 94 % of the data set the visible cameras are consistent to within $\pm 1$ okta ($\pm 0.125$) in the cloud fraction detection, so that they can be used to validate the fractional cloud coverage determined by the IRCCAM. For this comparison, a data set of 242,249 images (Mobotix) and a data set of 184,746 images (Schreder) are available. This comparison is only performed for daytime data of the IRCCAM, because from the visible cameras only daytime data are available. The residuals and some statistical values of the differences between the IRCCAM and the visible cameras are shown in Figure 9 and Table 2. With a median value of 0.01, there is no considerable difference between the cloud fraction determined by the IRCCAM and the cloud fraction determined by the Mobotix camera. The differences between the IRCCAM and the Schreder algorithms are only slightly larger, with median values of 0.04 and 0.07 for $Schreder_{pmod}$ and Schreder respectively. Thus the IRCCAM is only marginally overestimating the cloud fraction in comparison to the cloud fraction determined by the visible cameras. The distributions of the residuals IRCCAM-Schreder and IRCCAM-$Schreder_{pmod}$ are quite symmetrical (Figure 9b and 9c). The distribution of the residuals in cloud fraction IRCCAM-Mobotix is slightly left-skewed.

The percentage of agreement in the determined cloud fraction between the sky cameras and APCADA separately is given in Table 3. All values above the grey cells designate the fraction of data that agree within $\pm 0.125$ ($\pm 1$ okta) fractional cloud coverage between two individual algorithms and all values below the grey cells indicate the fraction that agree within $\pm 0.25$ ($\pm 2$ oktas) cloud fraction. The agreement of the IRCCAM in comparison with different visible all-sky cameras and APCADA is that 59–77 % of the IRCCAM data are within $\pm$ 0.125 ($\pm 1$ okta) fractional cloud coverage and 78 - 93 % of the data are within $\pm 0.25$ ($\pm 2$ oktas) fractional cloud coverage. We can conclude that the IRCCAM retrieves cloud fraction values within the uncertainty range of the cloud fraction retrieved from the visible cameras and also in a similar range as state of the art cloud detection instruments. These values of the IRCCAM are only slightly lower than the agreement that the visible cameras have amongst each other (94 - 100 % and 77 - 94 % are within $\pm 2$ oktas and $\pm 1$ okta respectively). The close agreement between the two algorithms Schreder and $Schreder_{pmod}$ is noteworthy, although they analyse a different number of pixels of the images.

### 3.2.1 Cloud Class Analysis

Although the median difference between the cloud fraction determined with the IRCCAM algorithm and the cloud fraction determined with the Mobotix algorithm is not evident, it is interesting to analyse differences in cloud fraction depending on the cloud type. The algorithm developed by Wacker et al. (2015) is used to distinguish six selected cloud classes and cloud-free cases automatically on the basis of the Mobotix images. Figure 10 shows the distribution of the residuals of the cloud fraction of the two aforementioned algorithms for (a) cumulus (low-level; N=37,320), (b) cirrocumulus-altocumulus (mid-level; N=52,097) and (c) cirrus-cirrostratus (high-level; N=10,467). The median value of the difference in cloud fraction between IRCCAM and Mobotix for Cu clouds is 0.02 and therefore not considerable. In general, all low-level clouds (Sc, Cu, St-As, Cb-Ns) are detected with a median cloud fraction difference of - 0.01 to 0.02 (Table 4). The IRCCAM and the Mobotix camera observe the mid-level cloud class Cc-Ac with a median agreement of cloud fraction of 0.00, but with a slightly asymmetric

distribution towards negative values. Considering 90 % of the data set of Cc-Ac clouds, the IRCCAM tends to underestimate the cloud fraction for the mid-level cloud class. The spread in the Cc-Ac data (shown as 5th and 95th percentiles in Table 4) is in general slightly larger than that for low-level clouds. The median value of the cloud fraction residuals determined on the basis of IRCCAM images and those based on Mobotix images for the high-level cloud class Ci-Cs is, at -0.13, clearly larger

in comparison to clouds at lower levels. Thus, although we applied the second part of the algorithm to detect thin, high-level clouds from the IRCCAM images, it still misses a large fraction of the Ci-Cs clouds in comparison to the Mobotix camera. The distribution of the residuals (Figure 10c) is clearly wider, which leads to 5th and 95th percentiles of -0.42 and 0.21 respectively. Due to the large spread, and also as shown in Aebi et al. (2017), the visible camera systems also have difficulties in detecting the thin, high-level clouds.

### 3.2.2   Day-night differences

So far, only daytime data have been analysed. At PMOD/WRC in Davos, during nighttime the cloud fraction is retrieved from pyrgeometers as well as from the IRCCAM. Therefore the IRCCAM cloud coverage data are compared with the data retrieved from the automated partial cloud amount detection algorithm (APCADA), which uses pyrgeometer data and calculates cloud

fractions independent of the time of day. As explained in Section 2.4, APCADA only determines the cloud fraction from low- to mid-level clouds and gives no information about high-level clouds. It also gives the cloud fraction only in okta-steps (equals steps of 0.125 cloud fraction).

Table 5 shows the median values of the residuals of the cloud fraction between IRCCAM and APCADA for all available data (N=103,635), only daytime data (N=32,902) and only nighttime data (N=70,722) and the corresponding 5th and 95th

percentiles separately. The overall median difference value in cloud fraction detection between IRCCAM and APCADA is, at 0.05, in a similar range as the ones for the comparison of the cloud fraction determined with the cloud cameras. The median value for daytime data is, at 0.06, only slightly larger than the one for nighttime data (0.04). However, the spread of the residuals is notably broad mainly during nighttime with a large positive 95th percentile value (0.65). However, because APCADA already showed larger spreads in the residuals in comparison to the fractional cloud coverage determined with the visible all-

sky cameras, it is not possible to draw the conclusion that the IRCCAM is overestimating the cloud fraction at nighttime.

### 3.2.3   Seasonal variations

The seasonal analysis is performed in order to investigate whether a slightly unequal distribution of cloud types in different months in Davos (Aebi et al., 2017) have an impact on the performance of the cloud fraction retrieval between seasons. The

percentage of agreement in the retrieved cloud fraction between the systems is again given for maximum $\pm 1$ okta ($\pm 0.125$) differences (top) and $\pm 2$ oktas ($\pm 0.25$) differences (bottom) for summer (left values) and winter (right values) in Table 6. For all algorithms there is a slightly closer agreement in the determined cloud fraction in the winter months in comparison to the summer months. In winter, the IRCCAM agrees with the other cameras in 78 - 83 % of the data within $\pm 0.125$ ($\pm 1$ okta)

and as high as 84 - 94 % within ±0.25 (±2 oktas). In summer, the agreement in cloud fraction is only 54 - 71 % of the data within ±0.125 (±1 okta) cloud fraction, but nevertheless, 84 - 91 % of values fall within ±0.25 (±2 oktas) cloud fraction. The slight difference between the two seasons might be explained by the slightly larger frequency of occurrence of the thin and low-emissivity cloud class cirrocumulus-altocumulus in Davos in summer than in winter (Aebi et al., 2017). Also the values
for spring (MAM) and autumn (SON) are in a similar range as the ones for summer and winter. Thus, the IRCCAM (and also the other camera systems) do not show any noteworthy variation in any of the seasons.

## 4   Conclusions

The current study describes a newly developed instrument - the thermal infrared cloud camera (IRCCAM) and its algorithm -
to determine cloud fraction on the basis of absolute sky radiance distributions. The cloud fraction determined on the basis of IRCCAM images is compared with the cloud fraction determined on the basis of images from two visible camera systems (one analysed with two different algorithms) and with the partial cloud amount determined with APCADA.

The overall median differences between the determined cloud fraction from the IRCCAM and the fractional cloud coverage determined from other instruments and algorithms are 0.01 - 0.07 fractional cloud coverage. The IRCCAM has an agreement of
±2 oktas (±0.25) in more than 90 % of cases and an agreement of ±1 okta (±0.125) in up to 77 % of the cases in comparison to other instruments. Thus, in only 10 % of the data, the IRCCAM typically overestimates the cloud fraction in comparison with the cloud fraction determined from the all-sky cameras sensitive in the visible region of the spectrum. Differences in the cloud fraction estimates can be due to different thresholds for the camera systems (as discussed in Calbo et al. (2017)) as well as some other issues addressed throughout the current study.

In general, there is no considerable difference in the performance of the IRCCAM in the different seasons. Analysis of the median values of the residuals between the cloud fraction determined from the IRCCAM with the ones calculated from APCADA shows no difference between day and night time, even though the spread of the residuals is clearly higher during nighttime.

The cloud fraction determination of the three cameras is independent of cloud classes, with the exception of thin cirrus clouds which are underestimated by the current IRCCAM algorithm by about 0.13 cloud fraction.

Overall, the IRCCAM is able to determine cloud fraction with a good agreement in comparison to all-sky cameras sensitive in the visible spectrum and with no considerable differences in its performance during different times of the day or different seasons. Thus, the IRCCAM is a stable system that can be used 24 hours per day with a high temporal resolution. In comparison to other state of the art cloud detection instruments (e.g. ceilometer or Nubiscope) it has the advantage of measuring the whole upper hemisphere at one specific moment. Its accuracy ranges from similar to rather better than that of the Nubiscope
(Feister et al., 2010) as well as that of the human observers (Boers et al., 2010).

In this study we mainly showed one application of the IRCCAM, which is to retrieve fractional cloud coverage information from the images. However, the known brightness temperature distribution of the sky and thus the known radiance can also be used for other applications including the determination of other cloud parameters (cloud type, cloud level, cloud optical thick-

ness) as well as the retrieval of information about downward longwave radiation in general. Thus, after some improvements in the hardware (e.g. a heating or ventilation system to avoid a frozen mirror) and software (improvements of the cloud algorithm detecting low-emissivity clouds by e.g. pattern recognition) the IRCCAM might be of interest for a number of further applications for example at meteorological stations or airports.

*Competing interests.* The authors declare that they have no conflict of interest.

*Acknowledgements.* This research was carried out within the framework of the project *A Comprehensive Radiation Flux Assessment (CRUX)* financed by MeteoSwiss. The project was funded in autumn 2013.

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

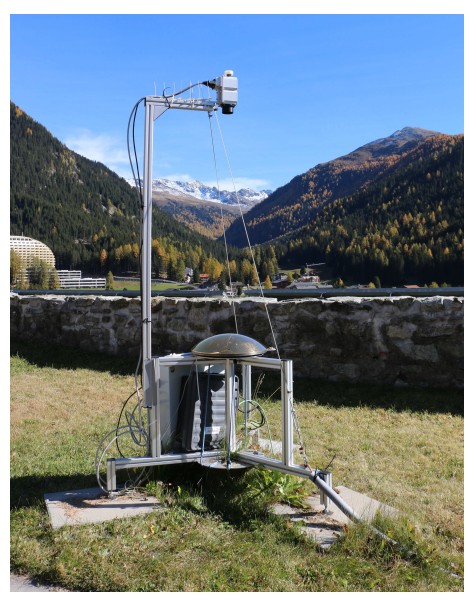

**Figure 1.** The Infrared Cloud Camera (IRCCAM) in the measurement enclosure of PMOD/WRC in Davos, Switzerland.

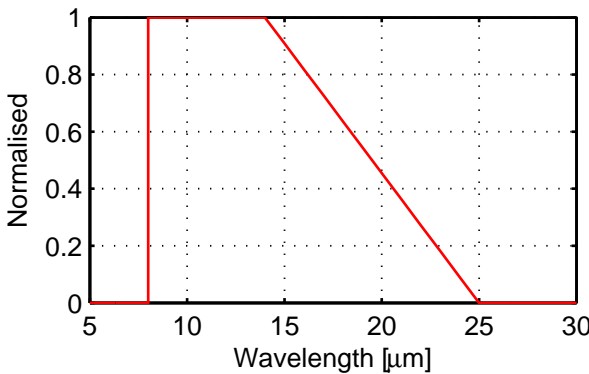

**Figure 2.** Response function $R_\lambda$ of the camera of the IRCCAM instrument.

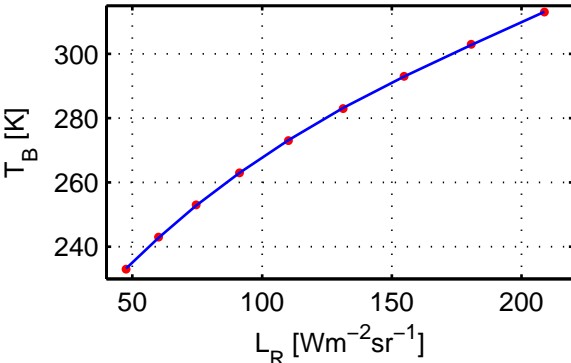

**Figure 3.** Brightness temperature $T_B$ versus integrated radiance $L_R$ for different radiance values (red dots), and the corresponding third order polynomial fitting function (blue line).

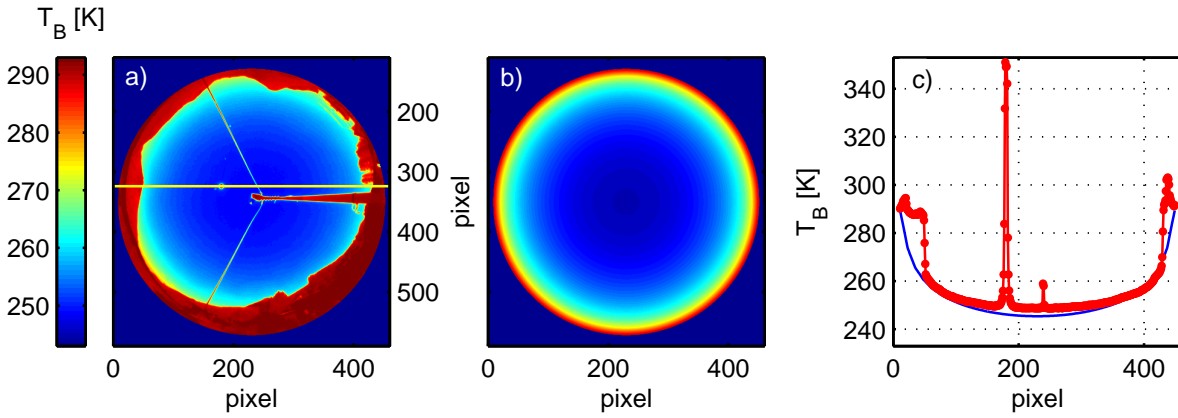

**Figure 4.** (a) Measured brightness temperature on the cloud-free day June 18, 2017 10:49 UTC (SZA=24 °), (b) the corresponding modelled brightness temperature and (c) the measured (red) and modelled (blue) profile of the sky brightness temperature along one azimuth position (shown as a yellow line in (a)).

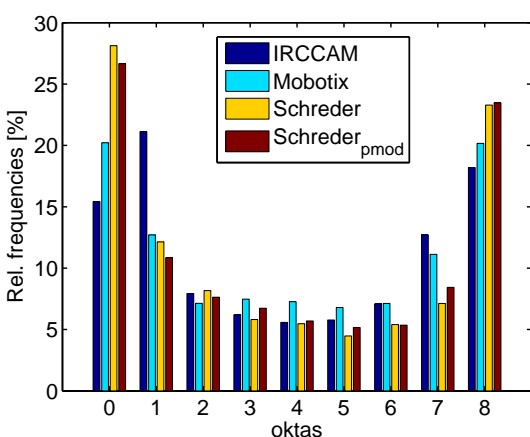

**Figure 5.** Relative frequencies of the determined cloud coverage of the study instruments for selected bins of cloud coverages at Davos. Zero okta: 0 - 0.0500; 1 okta: 0.0500 - 0.1875; 2 oktas: 0.1875 - 0.3125; 3 oktas: 0.3125 - 0.4375; 4 oktas: 0.4375 - 0.5625; 5 oktas: 0.5625 - 0.6875; 6 oktas: 0.6875 - 0.8125; 7 oktas: 0.8125 - 0.9500; 8 oktas: 0.9500 - 1;

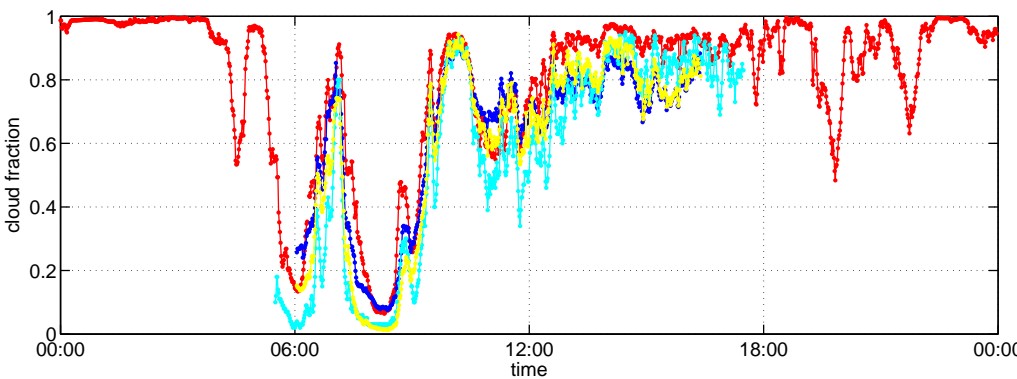

**Figure 6.** Cloud fraction determined by the analysed cameras and algorithms (red: IRCCAM, blue: Mobotix, cyan: Schreder, yellow: Schreder$_{pmod}$) on April 4, 2016.

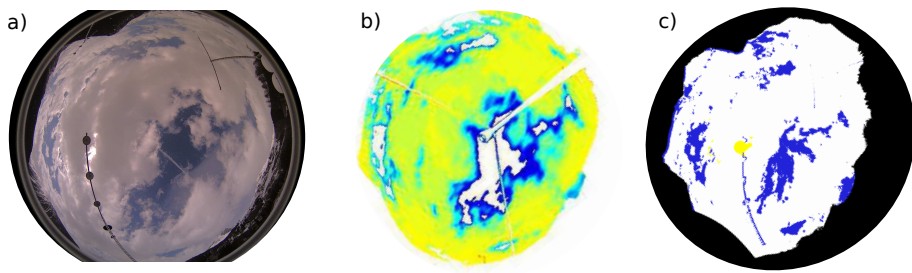

**Figure 7.** The cloud situation on April 4, 2016 10 UTC on an image from Mobotix (a) and the cloud fraction determined from (b) IRCCAM (temperature range from 244 K (blue) to 274 K (yellow)) and (c) Mobotix (white: clouds, blue: cloud-free, yellow: area around sun).

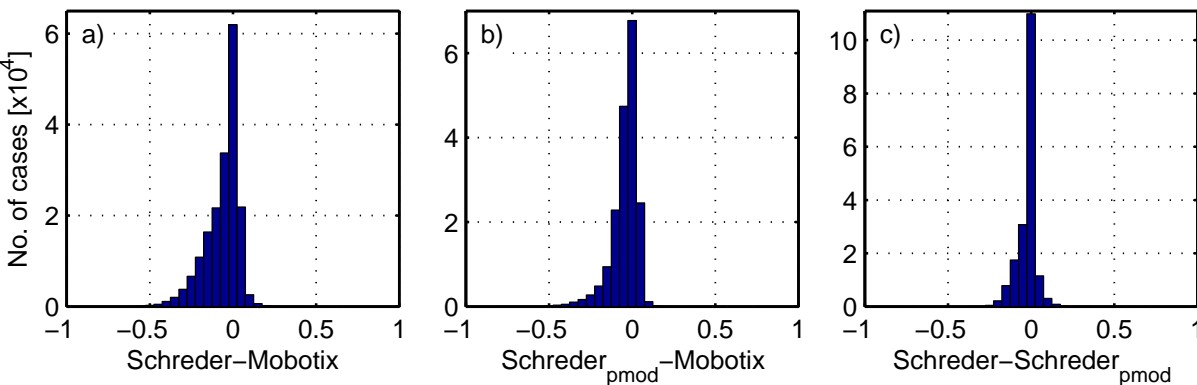

**Figure 8.** Residuals of the comparison of cloud fraction retrieved from the visible cameras and algorithms used in the study.

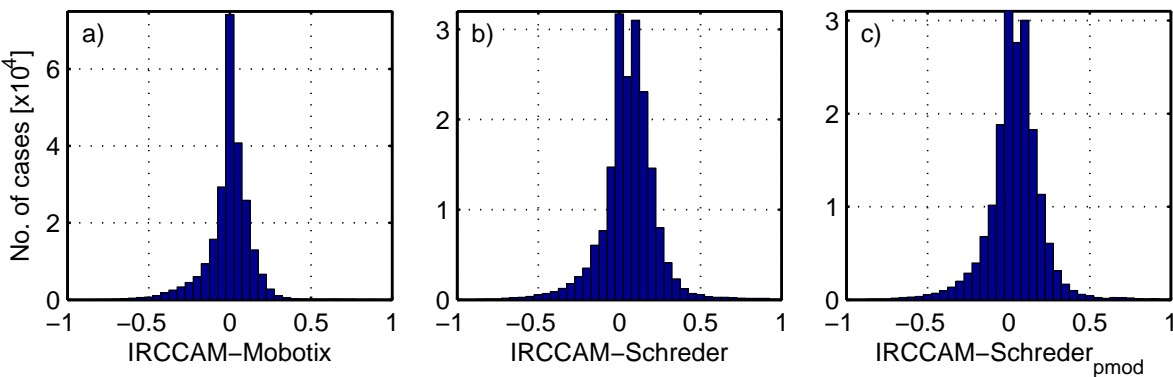

**Figure 9.** Residuals of the comparison of cloud fraction retrieved from the IRCCAM versus cloud fraction retrieved from the visible cameras.

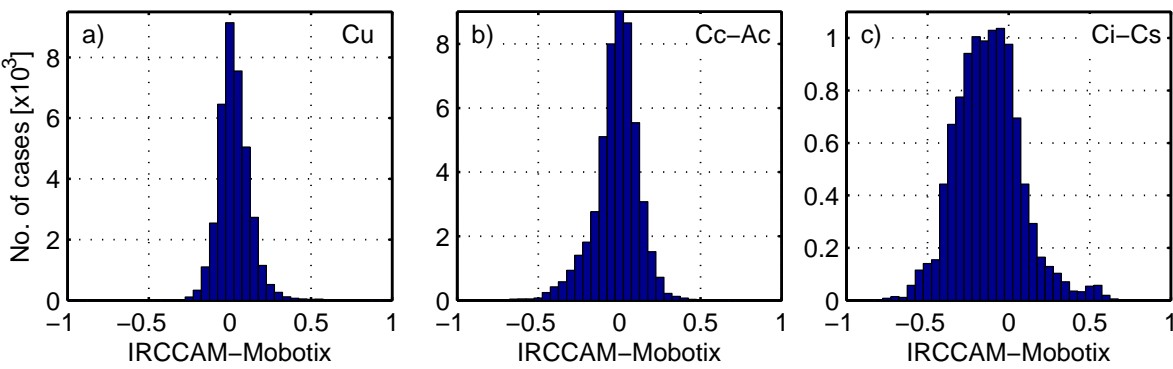

**Figure 10.** Residuals of the comparison of cloud fraction determined from IRCCAM images versus cloud fraction determined from Mobotix images for the following cloud classes: (a) Cu: Cumulus, (b) Cc-Ac: Cirrocumulus-Altocumulus and (c) Ci-Cs: Cirrus-Cirrostratus.

**Table 1.** Median and 5th and 95th percentiles of the differences in calculated cloud fractions from the visible all-sky cameras and APCADA. The numbers are in the range [-1;1].

| | Cloud fraction | | |
|---|---|---|---|
| | median | 5th | 95th |
| Schreder - Mobotix | -0.03 | -0.26 | 0.05 |
| Schreder$_{pmod}$ - Mobotix | -0.02 | -0.19 | 0.04 |
| Schreder - Schreder$_{pmod}$ | 0.00 | -0.13 | 0.04 |
| APCADA - Mobotix | -0.04 | -0.43 | 0.17 |
| APCADA - Schreder | -0.01 | -0.38 | 0.30 |
| APCADA - Schreder$_{pmod}$ | -0.01 | -0.38 | 0.26 |

**Table 2.** Median and 5th and 95th percentiles of the differences in calculated cloud fractions between IRCCAM and the visible all-sky cameras. The numbers are in the range [-1;1].

|  | Cloud fraction | | |
| --- | --- | --- | --- |
|  | median | 5th | 95th |
| IRCCAM - Mobotix | 0.01 | -0.26 | 0.18 |
| IRCCAM - Schreder | 0.07 | -0.22 | 0.29 |
| IRCCAM - Schreder$_{pmod}$ | 0.04 | -0.23 | 0.26 |

**Table 3.** Percentage of fractional cloud coverage data which agree within $\pm 1$ okta (all values above the grey cells) and $\pm 2$ oktas (all values below the grey cells) when comparing two algorithms each.

| | IRCCAM | Mobotix | Schreder | Schreder$_{pmod}$ | APCADA |
|---|---|---|---|---|---|
| IRCCAM | - | 77% | 59% | 66% | 62% |
| Mobotix | 93% | - | 77% | 89% | 67% |
| Schreder | 88% | 94% | - | 94% | 71% |
| Schreder$_{pmod}$ | 90% | 97% | 100% | - | 70% |
| APCADA | 80% | 83% | 86% | 85% | - |

**Table 4.** Median and 5th and 95th percentiles of the differences in calculated cloud fractions from IRCCAM and Mobotix images for selected cloud classes (stratocumulus (Sc), cumulus (Cu), stratus-altostratus (St-As), cumulonimbus-nimbostratus (Cb-Ns), cirrocumulus-altocumulus (Cc-Ac), cirrus-cirrostratus (Ci-Cs) and cloud-free (Cf). The numbers are in the range [-1;1].

|       | Cloud fraction | | |
| --- | --- | --- | --- |
|       | median | 5th | 95th |
| Sc    | 0.01   | -0.24 | 0.21 |
| Cu    | 0.02   | -0.12 | 0.19 |
| St-As | 0.00   | -0.38 | 0.11 |
| Cb-Ns | -0.01  | -0.22 | 0.08 |
| Cc-Ac | 0.00   | -0.27 | 0.18 |
| Ci-Cs | -0.13  | -0.42 | 0.21 |
| Cf    | 0.03   | -0.03 | 0.18 |

**Table 5.** Median and 5th and 95th percentiles of the differences in calculated cloud fractions from IRCCAM versus APCADA: overall; only daytime and only nighttime. The numbers are in the range [-1;1].

|  | Cloud fraction | | |
|---|---|---|---|
|  | median | 5th | 95th |
| IRCCAM - APCADA | 0.05 | -0.31 | 0.54 |
| IRCCAM - APCADA day | 0.06 | -0.18 | 0.35 |
| IRCCAM - APCADA night | 0.04 | -0.40 | 0.65 |

**Table 6.** Identical to Table 3, but left-hand are the values for the summer months (June, July, August) and right-hand the values for the winter months (December, January, February).

|  | IRCCAM | Mobotix | Schreder | Schreder$_{pmod}$ | APCADA |
|---|---|---|---|---|---|
| IRCCAM | - | 71% \| 83% | 54% \| 78% | 61% \| 80% | 62% \| 51% |
| Mobotix | 91% \| 94% | - | 76% \| 84% | 90% \| 87% | 66% \| 74% |
| Schreder | 89% \| 84% | 95% \| 93% | - | 93% \| 97% | 73% \| 89% |
| Schreder$_{pmod}$ | 89% \| 86% | 98% \| 95% | 100% \| 100% | - | 71% \| 92% |
| APCADA | 87% \| 65% | 84% \| 87% | 90% \| 97% | 88% \| 98% | - |