# Peer review of "Cloud fraction determined by thermal infrared and visible all-sky cameras"

_Atmospheric Measurement Techniques, 2018_

## Referee Comment (RC1) · P. Kuhn (Referee) · 13 Apr 2018

P. Kuhn (Referee)

pascal.kuhn@dlr.de

Summary

This script is concerned with an interesting and important field of research and should be published once major improvements are included.

Major comments:

1. I somewhat feel that the title could be more concise: Maybe you could add the word "comparison" and state the names of the used cameras.

2. Please discuss weaknesses / challenges of each studied system. How do the accuracies depend on (high) Linke turbidities, (low) solar angles or a "wet" atmosphere?

What other situations could lead higher deviations? This could be an own section (for each system or combined). Please discuss this quantitatively, with plots and figures.

3. Please add another section or at least a distinct paragraph in the introduction focused on the discussion of satellite cloud products and ground-based cameras. There is a Himawari-8 satellite, apparently with a cloud product down to 250 m and a sampling rate down to 2.5 min. The competition for ground-based cameras may not be human observers, but such satellites (see also minor comment 7). Where do you see the application of your cameras? What advantages do you see in comparison to satellites? This discussion could include the silhouette effect and projection uncertainties relevant to ground-based point-like observers, but not present for satellites.

4. The challenges being present regarding human cloud observation are partially addressed. However, you might be able to enhance the discussion. What is "not objective" (p1, line 24)? What are the differences if several experts evaluate images? You might be able to find such figures, which would significantly increase the quality of this argument. There is a reference missing on p2, line 1, for "nighttime determinations are difficult" - again, if you could find a reference, this would be an improvement.

5. Similar to major comment 3: If you state that "thin clouds cannot be distinguished from land" (p2, l9), you might as well enter into a full scale discussion of the weaknesses of satellites (beneficial to the quality of this paper). Please clarify the statement. Similar: "collect mainly from the highest cloud levels" (p2, l11) – Can't satellites, to some extent, differentiate? What are these limitations / what is the advantage of ground-based cameras? Same argument hold for the next sentence: "in order to retrieve. . ." – these statements are very absolute. How good are the cloud products of satellites, having multiple channels / sensors? To my knowledge, thin ice clouds can be differentiated from cumulus clouds quite well.

6. Maybe, there are more ways to determine cloud coverages, which were not mentioned in the introduction. For instance, cloud coverages could be estimated from PV-data or using downward-facing cameras (https://doi.org/10.1016/j.solener.2017.05.074, https://doi.org/10.5194/asr-15-11-2018 ).

7. Please provide a cost estimate of all used systems.

8. P2, l. 28f: I'm not an expert on this, but radars (e.g. those rain radars in airplanes) have a "lack of information about the whole sky"?

9. P2, l. 30: Please clarify how a point-like measurement system such as a ceilometer can "detect", "with considerable accuracy", "a fully covered or cloud-free" situation. Isn't this just an assumption that the small measured cone is representative of the whole sky? Stationary clouds outside this cone would not be detected by the ceilometer.

10. P2, l. 33: "The most common all-sky cameras are the total sky imager" – could you provide some figures on that? How many systems has Reuniwatt (or other companies) sold? Given the known issues with the TSI, one of which is its age, this might not be a very good reference.

11. Update your references. You quote many old papers in an area of active research.

12. P. 3, l. 3: "low cost commercial cameras" "give no information during nighttime. There are several commercially available IR surveillance cameras out there. Are you sure that they are not used in meteorology as of today?

13. P. 3, l. 7ff: I think there are more IR systems, e.g. the Reuniwatt one. Maybe you can highlight the differences a bit more and improve the presentation (it is hard to get all the ranges out of the text).

14. Introduction: Please generally improve the readability and the structure of the introduction. Maybe sub-titles might help.

15. End of introduction: Clearly state your motivation to develop a new system? What are the advantages in comparison to other developments?

16. P4, l. 22: Is there any reason why you assume a flat response curve? For cameras in the visible spectrum, the curve is very far from being flat. Isn't there a data sheet available?

17. P5, l.1: I think I've just missed it: What does this calibration function include? Both the mirror deviations and potential deviations of the imaging system of the camera?

18. P5, l. 9: To my experience, calibrations (with cameras in the visible spectrum) conducted with the sun show relatively large deviations (as the sun disk is usually quite large). This is less of an issue for IR cameras. However, I wonder: Why did you choose the sun instead of the full moon or stars? Could you estimate the deviations (presumably very relevant for the algorithms in the circumsolar area) for this EOR?

19. Figure 3 c: Please provide a scatter density plot over all days similar to this figure. There seems to be an offset in the center, which might be better visible or disappear if more data are studied.

20. P6, l. 1: I was wondering about the mirror temperature and potential asymmetries. Could you briefly state if the one-sided heating of the sun leads to a temperature distribution on that mirror within your stated 1 K range? In Fig. 1, a wall is visible close to the IR-camera – is there a problem with radiated heat, e.g. during night-times? Could you briefly state something on the interplay between the ground temperature and that mirror? Do you expect aging effects on the mirror? How bad is the soling?

21. P6, l 7f: You are stating absolute values here, saying that there are no differences between night and day data. Does this also hold for relative figures (I assume that the temperatures at night are lower)?

22. P6, l. 18: I somewhat doubt if the "observed discrepancy of 4 K" is only caused by model parameters (which one? Your LUT?). Please make this discussion a bit more wholesome. Other attributing factors might be camera instabilities and maybe the effects named in major comment 20.

23. P6, l. 29: Please further motivate the threshold of 6.5 K. This might be done with an example image, including clouds. Was this threshold somewhat fitted to the data?

24. Section 2.1.1 – please enhance visualization, e.g. using a flow-chart or pseudo-code.

25. P7, l. 4: I'm wondering how big intra-cloud temperature variations are. Could it be that parts of the cloud are detected as such while some pixels within are below the thresholds? If so, algorithms such as region growing or compression based approaches might enhance the segmentation.

26. P7, l. 6: If possible, further motivate the threshold of 1.2 K – is there a physical explanation?

27. P7, l. 11: Quantify "usually". Elaborate on the whole paragraph (this corresponds to a general discussion of challenging situations and weaknesses of this device and does not have to be done at this position. I suggest dedicating a whole section to this discussion).

28. P7, l. 30: Why did you choose a custom resolution for the Mobotix camera? To my experience, 1/500 is a very bright exposure time (this might be solved by blocking out the sun, but I'm curious) – is this an issue? You are using ratios to segment clouds, why did you not use an automatic exposure time?

29. P8, l 4.: Please provide a brief statement on how good the Mobotix system performs under high turbidity conditions, using a simple threshold-based approach, as well as for low solar elevations.

30. Specify the total run time of each algorithm.

31. P8, l 19: 70° - isn't it quite a problem if the FOV of all systems is not the same? Cloud coverages might be correctly detected but yet different. The same holds for different occlusions.

32. Specify the distance between the cameras, provide example images for all.

33. Section 3.2: Aggregating the figures to 1/8-bins clearly reduces the deviations between the systems. This is good for many applications in nowadays meteorology. However, I wonder how much the deviations increase if the images are compared pixel-wise. Could you provide these figures?

34. Section 3.2.3: Please motivate why you assumed seasonal differences – what are the origins of the deviations? It is presumably not earth's inclination towards the sun. Identify these parameters and study them separately. Example: I could imagine that e.g. a wet atmosphere poses challenges for the IR system. A wet atmosphere can happen both during winter and during summer times (with different probabilities). Aggregating over many different conditions might make analyses more difficult.

35. You could separate a new section "Next steps" (or similar) from the conclusion, stating in more detail what could be made to further improve the system and why you think this would lead to more accurate results.

36. Figure 5.: There is an interesting offset visible at around 7.00 (please state time-zone, UTC+0?) – how comes? Is this caused by a different FOV or different occlusion affecting the systems?

37. Figure 6.: Please add a colorbar for Fig. 6b. Did you bother to mask out the camera arm and the suspension? The forest to the right is very finely masked out – couldn't there be minor issues due to moving trees?

38. Figure 7.: I might have missed it, but is there so far a discussion on this bias included in the script? (also Fig. 9). Could you provide the same plots for a Mobotix-Schreder comparison? This would help to evaluate the references.

39. In general: You compare cloud fraction estimations from different systems, all of which are not completely accepted by everyone in the community. Potentially, it could further strengthen your line of argumentation if you included a comparison against a

more established approach. This could be (1) satellites or (2) a comparison to a clear sky index derived from DNI measurements over the whole period (3), also unrealistic, from PV data or (4) from ceilometer data. Presumably, (2) is the way to do it.

40. Table 5.: 1 okta is quite a lot and I'm a bit concerned about the rather low values visible here (e.g. 59%). Maybe, looking at pixel-wise deviations (major comment 33) could cast a light on the origins of these rather large deviations. This is clearly as good as or even better than human observers, but I think satellite cloud products and other camera systems (e.g. https://doi.org/10.1002/pip.2968 or the works from Stefan Winkler) achieve smaller deviations.

41. Table 6.: You state that there are no significant deviations between the seasons. This is only partially backed by the figures in this table. Please clarify in greater detail.

Minor comments:

1. I'm just wondering: Is there no English name for "Physikalisch-Meteorologisches Observatorium Davos/..."?

2. In general, the language used could be a bit more fluent. Examples are "other study instruments" in the abstract ("other instruments used here") or "coverage of the sun with clouds" (p1, line 18).

3. A short summary of major comment no 2 (challenges) should find its way into the abstract.

4. I might have missed it, but why are there, in the abstract, two figures for low-level cloud, and one figure each for mid-level and high-level clouds?

5. Please rephrase the sentence between p1, line 17 and p1, line 19 (subpar English).

6. P1, line 20, the position of "globally" seems to be odd. I furthermore disagree with that statement - there are more cloud observations made by satellites than by human observers.

7. P1, line 21: Human observation has the "advantage" to be carried out "several times per day" - satellites have a higher sampling rate, what you also mention later.

8. P1, line 23: "there is no reference standard for human observers" - really? No manual from any organization?

9. P1, line 23: Humans are "independent of any technical failure" - please rephrase.

10. P2, l. 2, leave out "measurement"

11. P2, l. 2., This sentence could be rephrased to something like "Recent research has therefore been conducted to find automated cloud detection instruments to ..." (more concise English). You might mention the DWD objective to automate its stations in the next years.

12. P2, l.5, is "synoptic" the correct word here? What do you want to say?

13. P2, l. 5. "time resolution of 15 min", there is a rapid scan method with 5 min

14. P2, l. 11, "Earth" could be written "earth".

15. P2, l. 13, you measure the cloud coverage, not the cloud in general, "cloud measurement techniques".

16. P.2, l. 14: maybe "certain" instead of "different" (from what?). Also "radiometers" – do you mean scanning radiometers? For the clear-sky algorithms based on GHI and DHI measurements?

17. P2, l. 25: You use "reflected" and "scattered" in a very similar way. Maybe "backscattered" is better suited?

18. P3, l. 3: maybe "development" instead of "deployment"?

19. P3, l. 16: state also here which "commercial thermal camera" you use.

20. You might clarify the term FOV. Once it is used for the camera and once for the whole system. Is it really 180°?

21. P3, l. 27. Please clearly state also here the models of the used cameras.

22. P3, l. 28: "and a newly developed" -> "and the newly developed"

23. P4, l. 9, rephrase, there are too many "and"s

24. P4, l. 9, high -> large / tall

25. P5, l. 25f: Rephrase/shorten the sentence. You might try to shorten other sentences as well.

26. General: I think this is not your fault, but I'd prefer having the images directly in the text, not at the end of the script.

---

## Referee Comment (RC2) · J. Calbó (Referee) · 13 Apr 2018

This paper introduces a new sky camera, specifically an infrared camera which can take sky images both in daylight and nighttime conditions. The paper explains the algorithm that is applied to derive cloud cover from the images of this camera. Moreover, a thorough validation-comparison effort is performed between cloud cover derived from these images, from images of other two (visible, that is, only during daylight hours) commercial sky cameras, and from the APCADA algorithm (based on cloud effect on downward longwave radiation measured with a pyrgeometer).

In some way this paper is a follow-on of a previous paper by the same research group (Aebi et al, 2017, AMT) where they presented an analysis of a long series of diurnal

cloud cover obtained with a sky camera. The present paper, however, has several added values: the introduction of a new concept of an infrared sky camera (looking downwards to a convex mirror), the suggestion of the method for image processing, and the comparison with other estimations of cloud cover.

Therefore, the paper is worth of being published in AMT. It seems to me that a few changes could be considered to make it more complete and to get higher impact in the scientific community, but even in the present version, the paper may be good enough to merit publication.

Suggested general change:

- In order to make more significant the comparison among all estimations of cloud cover, authors could consider applying exactly the same horizon mask to all images. For example, they could use a mask for the part of the image that is below 70 deg. SZA (20 deg. over the "flat" horizon). In fact, even APCADA algorithm is unsensitive to clouds that are in the horizon, so using this mask for all images would make the comparison more homogeneous.

Minor changes and technical details to be corrected:

- The word "significant" is used several times in the manuscript. I have my doubts about this use, as no statistical tests are applied (at least, they are not mentioned). So I would suggest use "significant" with caution, as it has a meaning related to statistical tests. If possible, try to use another word. In page 11, line 25, it is said that a difference of 0.02 is statistically not significant, but with no reference to what statistical test is applied.

- In lines 13-24 tow different approaches for cloudiness estimation are summarized. But in my opinion they are not clearly differentiated. Calbó et al 2001 suggests a method based on pyranometer measurements (i.e., hemispheric measurement of solar irrandiance), which is very different to Nephelo or Nubiscope, which are measuring in the infrared and in a narrow field of view. Please consider slightly modifying the writing AMTD
of this paragraph.

- I wouldn't say that WSI is among the most common all-sky cameras (as it is indeed the TSI). The WSI is one of the pioneering cameras, and presents very interesting characteristics and developments, but, to my knowledge, is no usually commercialized and therefore, is not quite common.

- Eq (3). Could you explain why the zenith angle is divided by 65?

- It is interesting to note that different thresholds are used in the processing of the visible images. This could partly explain some of the difference found in this paper. In fact, selection of the threshold is critical to distinguish between a cloudy pixel, and a clear (but sometimes, containing aerosol) pixel. Some discussion on this matter may be found in Calbó et al 2017 (and other studies cited therein). [Calbó, J., C. N. Long, J. González, J. Augustine, and A. Mccomiskey, 2017: The thin border between cloud and aerosol: Sensitivity of several ground based observation techniques. Atmos. Res., 196, 248–260, doi:10.1016/j.atmosres.2017.06.010.]

- The authors recognize that IRCCAM fails at both extremes of the cloud cover distribution. This (unexpected) result should merit more attention, with a deeper discussion if possible.

- The first lines of section 3.2.3 (lines 21-29) do not address seasonal analyses, so I suggest moving them to another section.

- Somewhere in the Results or Conclusion sections, I would appreciate a short discussion of the present results in comparison with performance of other IR whole sky cameras (if you can find any) or other sky cameras that take night images. If no previous work can be found with an estimation of the performance of such night images, this should be highlighted in the paper. Suggested references: [Shields, J. E., M. E. Karr, R. W. Johnson, and A. R. Burden, 2013: Day/night whole sky imagers for 24-h cloud and sky assessment: history and overview. Appl. Opt., 52, 1605–
1616, doi:10.1364/AO.52.001605; Gacal, G.F.B. Antioquia, C., and N. Lagrosas, 2016: Ground-based detection of nighttime clouds using a digital camera. Appl. Opt., 55, 6040–6045, doi:10.1364/AO.55.006040.]

- Figure 4, caption and related text. There is some mistake in the definition of oktas from cloud fraction. According with the caption, 0 oktas is for cloud fraction between 0 and 0.05 (which looks correct to me) but 8 oktas is between 0.875 and 1.0, that is a much larger interval, which seems wrong (at least, it is not symmetrical). And, for example, 4 oktas should be 0.4375-0.5625 (that is, a bin centered in 0.5 with a width of 0.125). If you correct this, some differences among the methods may change.

- Table 1 and table 2 could be put together in a single matrix-like table (like the authors do in Table 5). In each cell (only in one triangle of the matrix) both the median and the percentiles may be written (for example, as 0.01 [-0.24,0.21].

---

## Referee Comment (RC3) · Anonymous Referee #3 · 2 May 2018

This manuscript introduces a new infrared sky camera and an applied cloud detection algorithm and a comparison with visible sky cameras. It represents a substantial contribution to scientific progress within the scope of AMT. The image processing method, based on down-welling longwave radiation, to estimate the amount of cloud cover is a unique approach, as is the determination of cloud type. I recommend that the manuscript be published, with consideration of the following comments.

1. Page 2, Line 33: The TSI is indeed probably the most common all-sky camera but the Solmirus ASIVA or Reuniwatt Sky InSight may currently be more common than the WSI.

2. Page 6, Line 23: A better description is needed for "IRCCAM frame". Does this include the camera, arm, and wire ropes?

3. Page 6, Line 20, Page 8, Line 2, and Page 8, Line 20: The horizon mask appears to be independently defined for each image and for each of the three cameras. Using the same horizon mask for all images would yield a better comparison.

---

## Author Comment (AC1) · 11 Jun 2018

**Reply to comments by P. Kuhn (Referee #1)**

on the manuscript "Cloud fraction determined by thermal infrared and visible all-sky cameras" by Aebi et al., submitted to Atmospheric Measurement Techniques.

We would like to thank the referee for the constructive comments that contributed to the improvement of the manuscript. Detailed answers to the comments are given below (bold: referee comment, regular font: author's response, italic: changes in the manuscript).

Summary
This script is concerned with an interesting and important field of research and should be published once major improvements are included.

Major comments:

1.  **I somewhat feel that the title could be more concise: Maybe you could add the word "comparison" and state the names of the used cameras.**

We valued your suggestion, but we think that the title is adequate to the content of the paper. We would also prefer to have the title as concise as possible.

2.  **Please discuss weaknesses / challenges of each studied system. How do the accuracies depend on (high) Linke turbidities, (low) solar angles or a "wet" atmosphere? What other situations could lead higher deviations? This could be an own section (for each system or combined). Please discuss this quantitatively, with plots and figures.**

Thanks for this comment. The authors are aware that at certain locations high turbidity situations, due to aerosols or water vapour, may lead to problems in analysing/interpreting the sky images. However, in Davos, throughout the year we measure rather low integrated water vapour (IWV) values (between 2 and 25 mm) and also low AOD values. Thus, for our study we cannot analyse the sensitivity of the cameras regarding these conditions.
Low solar angles can lead to a "whitening" of the images (as discussed in Long et al., 2006). Our Mobotix camera in Davos is installed on a solar tracker and the sun is shaded with a shading disk (as described on p. 8, l. 14f.). Therefore we do not have any problems with overexposed images due to low solar angles. Also with the Schreder camera we do not see any problems in situations with low solar angles.

3.  **Please add another section or at least a distinct paragraph in the introduction focused on the discussion of satellite cloud products and ground-based cameras. There is a Himawari-8 satellite, apparently with a cloud product down to 250 m and a sampling rate down to 2.5 min. The competition for ground-based cameras may not be human observers, but such satellites (see also minor comment 7). Where do you see the application of your cameras? What advantages do you see in comparison to satellites? This discussion could include the silhouette effect and projection uncertainties relevant to ground-based point-like observers, but not present for satellites.**

We increased the length of the introduction and extended the paragraph discussing satellite measurements (p. 2, l. 6ff.):

*An alternative to detect clouds from the ground by human observations is to detect them from space. With a temporal resolution of 5 to 15 minutes, Meteosat Second Generation (MSG) geostationary satellites are able to detect cloud coverage with a higher time resolution than is accomplished by human observers (Ricciardelli et al., 2010; Werkmeister et al., 2015). The geostationary satellite Himawari-8 (Da, 2015) even delivers cloud information with a temporal resolution of 2.5 to 10 minutes and a spacial resolution of 0.5 to 2 km. However, these geostationary satellites cover only a certain region of the globe. Circumpolar satellites (i.e. the MODIS satellites Terra and Aqua (Baum B.A., 2006; Ackerman et al., 2008)) determine cloud fraction globally, but for a specific region only four times a day. Satellites cover a larger area than ground-based instruments and are also able to deliver cloud information from regions where few ground-based instruments are available (e.g. in Arctic regions (Heymsfield et al., 2017) or over oceans). However, due to the large field of view (FOV) of satellites, small clouds can be overlooked (Ricciardelli et al., 2010). Another challenge with satellite data is the ability to distinguish thin clouds from land (Dybbroe et al., 2005; Ackerman et al., 2008). Furthermore, satellites collect information mainly from the highest cloud layer rather than the lower cloud layer closer to the earth's surface. Nowadays satellite data are validated and thus supported by ground-based cloud data. Different studies focusing on the comparison of the determined cloud fraction from ground and from space were presented by e.g. Fontana et al. (2013); Wacker et al. (2015); Calbo et al. (2016); Kotarba (2017).*

However, at this point we would like to mention, that the paper focuses on the description of a new ground-based instrument that might serve as an alternative to human cloud observations (as mentioned on p. 3, l. 33ff.) and does not focus on cloud observations from satellites.

The authors added some more possible applications for the newly developed IRCCAM (p. 3, l. 35/p. 4, l. 1.):

*Thus the IRCCAM could be used for different applications at meteorological stations, at airports or at solar power plants.*

4. The challenges being present regarding human cloud observation are partially addressed. However, you might be able to enhance the discussion. What is "not objective" (p1, line 24)? What are the differences if several experts evaluate images? You might be able to find such figures, which would significantly increase the quality of this argument. There is a reference missing on p2, line 1, for "nighttime determinations are difficult" - again, if you could find a reference, this would be an improvement.

We extended and changed the paragraph about human cloud observations (p. 1, l. 20ff.):

*The most common practice worldwide to determine cloud coverage, cloud base height (CBH) and cloud type from the ground are human observations (CIMO, 2014). These long-term series of cloud data allow climate studies to be conducted (e.g. Chernokulsky et al., 2017). Cloud detection by human observers is carried out several times per day over a long time period without the risk of a larger data gap due to a technical failure of an instrument. However, even with a reference standard defined by the World Meteorological Organisation (WMO) for human observers, the cloud determination is not objective e.g. mainly due to varying degrees of*

*experience (Boers et al., 2010). Other disadvantages of human cloud observations are that the temporal resolution is coarse and due to visibility issues nighttime determinations are difficult. Since clouds are highly variable in space and time, measurements at high spatial and temporal resolution with small uncertainties are needed (WMO, 2012). Recent research has therefore been conducted to find an automated cloud detection instrument (or a combination of such) to replace human observers (Boers et al., 2010; Tapakis and Charalambides, 2013; Huertas-Tato et al., 2017; Smith et al., 2017).*

5.  Similar to major comment 3: If you state that "thin clouds cannot be distinguished from land" (p2, l9), you might as well enter into a full scale discussion of the weaknesses of satellites (beneficial to the quality of this paper). Please clarify the statement. Similar: "collect mainly from the highest cloud levels" (p2, l11) – Can't satellites, to some extent, differentiate? What are these limitations / what is the advantage of ground-based cameras? Same argument hold for the next sentence: "in order to retrieve. . ." – these statements are very absolute. How good are the cloud products of satellites, having multiple channels / sensors? To my knowledge, thin ice clouds can be differentiated from cumulus clouds quite well.

As already mentioned in the answer to major comment 3, we extended the discussion about satellite cloud detection (p. 2, l. 6ff.). However, the authors would like to mention here again, that the main focus of the paper is on ground-based cloud detection and not on the cloud detection from satellites. Therefore, the authors think that it is not needed to go into further details about satellite experiments.

6.  Maybe, there are more ways to determine cloud coverages, which were not mentioned in the introduction. For instance, cloud coverages could be estimated from PV-data or using downward-facing cameras (https://doi.org/10.1016/j.solener.2017.05.074, **https://doi.org/10.5194/asr-15-11-2018** ).

Thanks for this comment, we included some more references in the text (for example p. 3, l. 13f.).

7.  Please provide a cost estimate of all used systems.

From the used camera systems the one from Mobotix is the least expensive. The price of the Schreder all-sky camera is in the order of five times as much as the Mobotix camera and the IRCCAM in the order of fifty times a Mobotix camera.

8.  P2, l. 28f: I'm not an expert on this, but radars (e.g. those rain radars in airplanes) have a "lack of information about the whole sky"?

The authors refer to cloud radar systems as for example described in Boers et al., 2010, which have a beam width of 0.3 degrees. Thus the cloud radars do also belong to the column cloud detection instruments.

9.  P2, l. 30: Please clarify how a point-like measurement system such as a ceilometer can "detect", "with considerable accuracy", "a fully covered or cloud-free" situation. Isn't this just an assumption that the small measured cone is representative of the whole sky? Stationary clouds outside this cone would not be detected by the ceilometer.

We revised our sentence (p. 3, l. 6f.):

*Boers et al. (2010) showed that with smaller integration times the instruments tend to give okta values of zero and eight rather than the intermediate cloud fractions of 1 to 7 oktas.*

10. P2, l. 33: "The most common all-sky cameras are the total sky imager" – could you provide some figures on that? How many systems has Reuniwatt (or other companies) sold? Given the known issues with the TSI, one of which is its age, this might not be a very good reference.

We are aware that the TSI is an older (and even one of the pioneering instruments), but it is still one of the most common all-sky cameras (as also mentioned by the other two referees).

11. Update your references. You quote many old papers in an area of active research.

We added some more recent publications in the text (for example p. 2, l. 5).

12. P. 3, l. 3: "low cost commercial cameras" "give no information during nighttime. There are several commercially available IR surveillance cameras out there. Are you sure that they are not used in meteorology as of today?

It is possible that some other commercially available IR surveillance cameras than the ones mentioned in the text are used in meteorology. However, at present we do not have any knowledge or did not find any publication about them.

13. P. 3, l. 7ff: I think there are more IR systems, e.g. the Reuniwatt one. Maybe you can highlight the differences a bit more and improve the presentation (it is hard to get all the ranges out of the text).

Thanks for this comment. As it seems the Sky Insight thermal infrared cloud imager from Reuniwatt is very new on the market and we were not aware of this instrument before submitting our discussion paper. In the revised version we added a few sentences about this camera system (p. 3, l. 26ff.):

*Relatively new on the market is the Sky Insight thermal infrared cloud imager from Reuniwatt. The Sky Insight cloud imager is sensitive in the 8 µm - 13 µm wavelength range and its layout and software is similar to the prototype instrument presented here.*

To our knowledge, there is no publication about the performance of this instrument available yet. Therefore we cannot highlight any differences in the performance.

We also added another recent publication about an infrared sky imaging system (p. 3, l. 25f.):

*Redman et al. (2018) presented a reflective all-sky imaging system (sensitive in the 8 µm - 14 µm wavelength range) consisting of a longwave infrared microbolometer camera and a reflective sphere (110° FOV).*

14. Introduction: Please generally improve the readability and the structure of the introduction. Maybe sub-titles might help.

The introduction was extended and rewritten following most of the referee's remarks.

15. End of introduction: Clearly state your motivation to develop a new system? What are the advantages in comparison to other developments?

We further motivated the development of a new camera system (p.3., l. 33ff.):

*The IRCCAM was developed to provide instantaneous hemispheric cloud coverage information from the ground with a high temporal resolution in a more objective way than human cloud observations. Thus the IRCCAM could be used for different applications at meteorological stations, at airports or at solar power plants.*

16. P4, l. 22: Is there any reason why you assume a flat response curve? For cameras in the visible spectrum, the curve is very far from being flat. Isn't there a data sheet available?

We actually did not assume a flat response function and considered the response function we received from the manufacturing company. We included Figure 2, which shows the actual response function $R_\lambda$. We changed Equation 2 correspondingly as well as the description in the text. The conclusions of our study does not change regardless of the definition of the response function.

17. P5, l.1: I think I've just missed it: What does this calibration function include? Both the mirror deviations and potential deviations of the imaging system of the camera?

The camera was placed in front of a blackbody aperture for retrieving the calibration function. This function is then independent from the mirror. The function's purpose is to convert the output of the camera (i.e. the number per pixel) to brightness temperature and radiance respectively.

18. P5, l. 9: To my experience, calibrations (with cameras in the visible spectrum) conducted with the sun show relatively large deviations (as the sun disk is usually quite large). This is less of an issue for IR cameras. However, I wonder: Why did you choose the sun instead of the full moon or stars? Could you estimate the deviations (presumably very relevant for the algorithms in the circumsolar area) for this EOR?

The solar disk on the image covers an area of around 1°. Thus we think that the sun covers a very well defined area on the images and we do not see any problem to use the sun to remove the distortion of the images. The full moon is only visible during cloud-free conditions and thus not practical to use it as a reference to undistort the images. Stars are not visible at all on the images.

19. Figure 3 c: Please provide a scatter density plot over all days similar to this figure. There seems to be an offset in the center, which might be better visible or disappear if more data are studied.

We are aware that there is an offset in the center of the images. Our analysis on p. 6, l. 28ff. is showing that there is an average difference between the measurement and the model of 4 K ±2.4 K which stems to a certain degree from this offset in the center. However, the authors think that this offset is not relevant for the present study to determine the cloud fraction.

20. P6, l. 1: I was wondering about the mirror temperature and potential asymmetries. Could you briefly state if the one-sided heating of the sun leads to a temperature distribution on that mirror within your stated 1 K range? In Fig. 1, a wall is visible close to the IR-camera – is there a problem with radiated heat, e.g. during night-times? Could you briefly state something on the interplay between the ground temperature and that mirror? Do you expect aging effects on the mirror? How bad is the soling?

We did not see any problems with asymmetries in the temperature distribution on the mirror. We also did not see any effect of the wall next to the IRCCAM on the sky brightness distribution on the mirror. However, what we have seen is the larger longwave emissivity from Davos (SSW direction), which leads to a false classification of cloudy pixels on the images in direction of Davos. This problem is briefly discussed on p. 10, l. 4ff.:

*It is noteworthy that the IRCCAM clearly underestimates the occurrence of 0 oktas in comparison to the cameras measuring in the visible spectrum (by up to 13 %). On the other hand, the relative frequency of the IRCCAM of 1 okta is clearly larger (by up to 10 %) compared to the visible cameras. This can be explained by higher brightness temperatures measured in the vicinity of the horizon above Davos. These higher measured brightness temperatures are falsely determined as cloudy pixels (up to 0.16 cloud fraction). Since these situations with larger brightness temperatures occur quite frequently, the IRCCAM algorithm detects more often cloud coverages of 1 okta instead of 0 okta.*

So far the mirror did not show any relevant aging issues.

21. P6, l 7f: You are stating absolute values here, saying that there are no differences between night and day data. Does this also hold for relative figures (I assume that the temperatures at night are lower)?

The absolute differences between night and day are 4.32 K ±2.3 K and 3.86 K ±2.5 K respectively. In relative numbers, the difference between day and night is around 0.2 % and thus negligible.

22. P6, l. 18: I somewhat doubt if the "observed discrepancy of 4 K" is only caused by model parameters (which one? Your LUT?). Please make this discussion a bit   more   wholesome. Other attributing factors might be camera instabilities and maybe the effects named in major comment 20.

We changed the sentence (p. 7, l. 1ff.):

*Therefore, the observed discrepancy of 4 K between measurements and model calculations mentioned previously can probably be attributed to the uncertainties in the model parameters (temperature and IWV) used to produce the LUT.*

**23. P6, l. 29: Please further motivate the threshold of 6.5 K. This might be done with an example image, including clouds. Was this threshold somewhat fitted to the data?**

The threshold of 6.5 K is empirically defined. We chose this rather large threshold to minimise the probability that cloud-free pixels are (wrongly) classified as clouds (described on p. 7, l. 16ff.). The second part of the algorithm decides whether thin (and therefore low-emissivity) clouds) are present or not.

**24. Section 2.1.1 – please enhance visualization, e.g. using a flow-chart or pseudocode.**

We slightly changed the description of the algorithm and are convinced that the changes increased the readability of this section.

**25. P7, l. 4: I'm wondering how big intra-cloud temperature variations are. Could it be that parts of the cloud are detected as such while some pixels within are below the thresholds? If so, algorithms such as region growing or compression based approaches might enhance the segmentation.**

What we see from the images is that there is a smooth decrease of the brightness temperature at the border of the clouds. Thus it becomes more difficult to detect certain pixels as clouds the smaller the difference in brightness temperature to cloud-free pixels is. This behaviour makes it also difficult to detect thin (low emissivity) clouds. In order to possibly improve the determination of thin cirrus clouds, a pattern recognition algorithm could be tested in a further study.

**26. P7, l. 6: If possible, further motivate the threshold of 1.2 K – is there a physical explanation?**

The threshold of 1.2 K is empirically defined. Different thresholds were tested for a certain number of images and thereafter we chose the threshold of 1.2 K because it was the best fit between classifying and mis-classifying cloudy/non-cloudy pixels.

**27. P7, l. 11: Quantify "usually". Elaborate on the whole paragraph (this corresponds to a general discussion of challenging situations and weaknesses of this device and does not have to be done at this position. I suggest dedicating a whole section to this discussion).**

We removed these sentences from this section.

**28. P7, l. 30: Why did you choose a custom resolution for the Mobotix camera? To my experience, 1/500 is a very bright exposure time (this might be solved by blocking out the sun, but I'm curious) – is this an issue? You are using ratios to segment clouds, why did you not use an automatic exposure time?**

Our Mobotix camera is installed on a solar tracker and is shaded with a shading disk (as mentioned on p. 8, l. 14f.) and the bright exposure time is therefore not an issue.

**29. P8, l. 4: Please provide a brief statement on how good the Mobotix system performs under high turbidity conditions, using a simple threshold-based approach, as well as for low solar elevations.**

See the answer to major comment 2.

**30. Specify the total run time of each algorithm.**

To calculate the cloud fraction from the two visible cameras for a full day takes on a personal computer a few minutes whereas the calculation of the cloud fraction from the IRCCAM takes 30-60 minutes.

**31. P8, l 19: 70° - isn't it quite a problem if the FOV of all systems is not the same? Cloud coverages might be correctly detected but yet different. The same holds for different occlusions.**

Indeed it is a problem if the FOV of all systems is not equal. This problem is also briefly discussed in different paragraphs of the paper (e.g. p. 11, l. 9ff.).

**32. Specify the distance between the cameras, provide example images for all.**

The distance between the Mobotix and the Schreder camera is roughly six meters. The IRCCAM is roughly 70 m and 76 m away from the Schreder and the Mobotix camera respectively.

**33. Section 3.2: Aggregating the figures to 1/8-bins clearly reduces the deviations between the systems. This is good for many applications in nowadays meteorology. However, I wonder how much the deviations increase if the images are compared pixelwise. Could you provide these figures?**

Thanks for this comment. It would be indeed interesting to compare the different cameras pixelwise, however, for the aim of our study, presenting a new camera system to detect clouds for synoptic purposes, this analysis is out of scope.

**34. Section 3.2.3: Please motivate why you assumed seasonal differences – what are the origins of the deviations? It is presumably not earth's inclination towards the sun. Identify these parameters and study them separately. Example: I could imagine that e.g. a wet atmosphere poses challenges for the IR system. A wet atmosphere can happen both during winter and during summer times (with different probabilities). Aggregating over many different conditions might make analyses more difficult.**

We added a motivation for the seasonal analysis (p. 13, l. 17ff.):

*The seasonal analysis is performed in order to investigate whether a slightly unequal distribution of cloud types in different months in Davos (Aebi et al., 2017) have an impact on the performance of the cloud fraction retrieval between seasons.*

**35. You could separate a new section "Next steps" (or similar) from the conclusion, stating in more detail what could be made to further improve the system and why you think this would lead to more accurate results.**

On p. 14, l. 26ff. we mentioned different points that could be tested in order to improve the system:

*However, the known brightness temperature distribution of the sky and thus the known radiance can also be used for other applications including the determination of other cloud parameters (cloud type, cloud level, cloud optical thickness) as well as the retrieval of information about downward longwave radiation in general. Thus, after some improvements in the hardware (e.g. a heating or ventilation system to avoid a frozen mirror) and software (improvements of the cloud algorithm detecting low-emissivity clouds by e.g. pattern recognition) the IRCCAM might be of interest for a number of further applications for example at meteorological stations or airports.*

**36. Figure 5.: There is an interesting offset visible at around 7.00 (please state timezone, UTC+0?) – how comes? Is this caused by a different FOV or different occlusion affecting the systems?**

Thanks for this comment. There were two time steps that had an extremely large cloud fraction determined by the IRCCAM. During those two time steps someone was cleaning the mirror with distilled water. A mirror covered with water leads to high emissivity values which are wrongly detected as clouds. Therefore we removed now these two data points from the Figure (now Figure 6).

**37. Figure 6.: Please add a colorbar for Fig. 6b. Did you bother to mask out the camera arm and the suspension? The forest to the right is very finely masked out – couldn't there be minor issues due to moving trees?**

We added the temperature range to the caption of the Figure. The features depicted on Figure 7c (before Figure 6c) are the shading disks of three sun trackers. Thus they are moving and therefore it is difficult to mask them out without losing information about the sky pixels. In Figure 7c (before Figure 6c), these shading disk covers an area of only 0.4 %. The area between the trees covers less than 0.01 % of the analysed pixels and is therefore not relevant for the current study, even in cases of moving trees.

**38. Figure 7.: I might have missed it, but is there so far a discussion on this bias included in the script? (also Fig. 9). Could you provide the same plots for a Mobotix-Schreder comparison? This would help to evaluate the references.**

The focus of our study is to present and validate the performance of the IRCCAM regarding cloud fraction determination. Therefore, the authors are convinced that it is not relevant to increase the discussion about the comparison between Mobotix and Schreder cameras.

**39. In general: You compare cloud fraction estimations from different systems, all of which are not completely accepted by everyone in the community. Potentially, it could further strengthen your line of argumentation if you included a comparison against a more established approach. This could be (1) satellites or (2) a comparison to a clear sky index derived from DNI measurements over the whole period (3), also unrealistic, from PV data or (4) from ceilometer data. Presumably, (2) is the way to do it.**

The Mobotix camera has been validated against and compared with data from satellites, human observers and a ceilometer in former studies (for example in Wacker et al., 2015). Those studies are a valid argument for using the Mobotix camera as a reference for the validation of the newly

developed IRCCAM. In Davos we do not have possibilities to compare our data with ceilometer or PV data. As mentioned in the paper as well as in several answers here, the study focuses on the cloud detection with ground-based instruments and not on comparisons with satellite data.

40. Table 5.: 1 okta is quite a lot and I'm a bit concerned about the rather low values visible here (e.g. 59%). Maybe, looking at pixel-wise deviations (major comment 33) could cast a light on the origins of these rather large deviations. This is clearly as good as or even better than human observers, but I think satellite cloud products and other camera systems (e.g. https://doi.org/10.1002/pip.2968 or the works from Stefan Winkler) achieve smaller deviations.

The focus of the paper is to present a newly developed ground based camera that might be used for example at meteorological stations, airports or solar power plants. At meteorological stations the state of the art unit for cloud fraction is oktas (also defined by WMO).
As we already mentioned in the major comment 3 and 5, the focus of the paper is on ground-based measurements and not on satellite data.

41. Table 6.: You state that there are no significant deviations between the seasons. This is only partially backed by the figures in this table. Please clarify in greater detail.

We extended the discussion in Section 3.2.3 (p. 13, l. 25ff.):

*The slight difference between the two seasons might be explained by the slightly larger frequency of occurrence of the thin and low-emissivity cloud class cirrocumulus-altocumulus in Davos in summer than in winter (Aebi et al., 2017).*

Minor comments:

1. I'm just wondering: Is there no English name for "Physikalisch-Meteorologisches Observatorium Davos/..."?

This is the official affiliation of our institute and is therefore not translated.

2. In general, the language used could be a bit more fluent. Examples are "other study instruments" in the abstract ("other instruments used here") or "coverage of the sun with clouds" (p1, line 18).

Done

3. A short summary of major comment no 2 (challenges) should find its way into the abstract.

See the answer to major comment number 2.

4. I might have missed it, but why are there, in the abstract, two figures for low-level cloud, and one figure each for mid-level and high-level clouds?

The abstract has been rewritten to a larger part.

5. Please rephrase the sentence between p1, line 17 and p1, line 19 (subpar English).

Done

6. P1, line 20, the position of "globally" seems to be odd. I furthermore disagree with that statement - there are more cloud observations made by satellites than by human observers.

Following most of your recommendations, we changed a large fraction of the introduction.

7. P1, line 21: Human observation has the "advantage" to be carried out "several times per day" - satellites have a higher sampling rate, what you also mention later.

Following most of your recommendations, we changed a large fraction of the introduction.

8. P1, line 23: "there is no reference standard for human observers" - really? No manual from any organization?

We rewrote the sentence on p. 1, l. 23ff.:

*However, even with a reference standard defined by the World Meteorological Organisation (WMO) for human observers, the cloud determination is not objective e.g. mainly due to varying degrees of experience (Boers et al., 2010).*

9. P1, line 23: Humans are "independent of any technical failure" - please rephrase.

Done

10. P2, l. 2, leave out "measurement"

Done

11. P2, l. 2, This sentence could be rephrased to something like "Recent research has therefore been conducted to find automated cloud detection instruments to ..." (more concise English). You might mention the DWD objective to automate its stations in the next years.

Following most of your recommendations, we changed a large fraction of the introduction.

12. P2, l.5, is "synoptic" the correct word here? What do you want to say?

Following most of your recommendations, we changed a large fraction of the introduction.

13. P2, l. 5. "time resolution of 15 min", there is a rapid scan method with 5 min

We changed the sentence on p. 2, l. 6ff.:

*With a temporal resolution of 5 to 15 minutes, Meteosat Second Generation (MSG) geostationary satellites are able to detect cloud coverage with a higher time resolution than it is accomplished by human observers (Ricciardelli et al., 2010; Werkmeister et al., 2015).*

14. P2, l. 11, "Earth" could be written "earth".

Done

15. P2, l. 13, you measure the cloud coverage, not the cloud in general, "cloud measurement techniques".

Done

16. P.2, l. 14: maybe "certain" instead of "different" (from what?). Also "radiometers" – do you mean scanning radiometers? For the clear-sky algorithms based on GHI and DHI measurements?

We changed the whole paragraph.

17. P2, l. 25: You use "reflected" and "scattered" in a very similar way. Maybe "backscattered" is better suited?

Done

18. P3, l. 3: maybe "development" instead of "deployment"?

Done

19. P3, l. 16: state also here which "commercial thermal camera" you use.

Done

20. You might clarify the term FOV. Once it is used for the camera and once for the whole system. Is it really 180°?

The IRCCAM has a field of view of 180°. But the effective view of the sky is defined by the horizon, which is in a mountainous area as Davos clearly less than the 180°.

21. P3, l. 27. Please clearly state also here the models of the used cameras.

Done

22. P3, l. 28: "and a newly developed" -> "and the newly developed"

Done

23. P4, l. 9, rephrase, there are too many "and"s

Done

24. P4, l. 9, high -> large / tall

Done

25. P5, l. 25f: Rephrase/shorten the sentence. You might try to shorten other sentences as well.

Done

26. General: I think this is not your fault, but I'd prefer having the images directly in the text, not at the end of the script.

This is indeed not the decision of the authors, but the guideline of the journal.

---

## Author Comment (AC3) · 11 Jun 2018

**Reply to comments by Anonymous Referee #3**

on the manuscript "Cloud fraction determined by thermal infrared and visible all-sky cameras" by Aebi et al., submitted to Atmospheric Measurement Techniques.

We would like to thank the referee for the constructive comments that contributed to the improvement of the manuscript. Detailed answers to the comments are given below (bold: referee comment, regular font: author's response, italic: changes in the manuscript).

**This manuscript introduces a new infrared sky camera and an applied cloud detection algorithm and a comparison with visible sky cameras. It represents a substantial contribution to scientific progress within the scope of AMT. The image processing method, based on down-welling longwave radiation, to estimate the amount of cloud cover is a unique approach, as is the determination of cloud type. I recommend that the manuscript be published, with consideration of the following comments.**

1. **Page 2, Line 33: The TSI is indeed probably the most common all-sky camera but the Solmirus ASIVA or Reuniwatt Sky InSight may currently be more common than the WSI.**

Thanks for this comment. We included a short description about the Reuinwatt Sky InSight cloud imager in our introduction (p. 3, l. 26ff.):

*Relatively new on the market is the Sky Insight thermal infrared cloud imager from Reuniwatt. The Sky Insight cloud imager is sensitive in the 8 μm - 13 μm wavelength range and its layout and software is similar to the prototype instrument presented here.*

We also slightly adapted the paragraphs discussing the TSI, WSI and the Solmirus ASIVA:

p. 3, l. 8f.:
*The most common all-sky camera is the commercially available total sky imager (TSI) (Long et al., 2006). Another pioneering hemispherical cloud detection instrument is the whole sky imager (WSI) (Shields et al., 2013).*

p. 3, l. 19ff.:
*Another instrument, the Solmirus all-sky infrared visible analyser (ASIVA) consists of two cameras, one measuring in the visible and the other one in the 8 μm - 13 μm wavelength range (Klebe et al., 2014).*

2. **Page 6, Line 23: A better description is needed for "IRCCAM frame". Does this include the camera, arm, and wire ropes?**

Yes, the term "IRCCAM frame" includes the camera, arm and wire ropes. We clarified this in the text (p. 7, l. 7ff.):

*This image mask contains local obstructions such as the IRCCAM frame (camera, arm and wire ropes) as well as the horizon, which in the case of Davos consists of mountains limiting the field of view of the IRCCAM.*

3. 3. Page 6, Line 20, Page 8, Line 2, and Page 8, Line 20: The horizon mask appears to be independently defined for each image and for each of the three cameras. Using the same horizon mask for all images would yield a better comparison.

There is only one horizon mask per camera (we clarified this now in the text). Because the resolution and the location of the three cameras is slightly different, we decided to define one horizon mask per camera system and not using the same for all systems.

---

## Author Comment (AC2)

**Reply to comments by J. Calbó (Referee #2)**

on the manuscript "Cloud fraction determined by thermal infrared and visible all-sky cameras" by Aebi et al., submitted to Atmospheric Measurement Techniques.

We would like to thank the referee for the constructive comments that contributed to the improvement of the manuscript. Detailed answers to the comments are given below (bold: referee comment, regular font: author's response, italic: changes in the manuscript).

**This paper introduces a new sky camera, specifically an infrared camera which can take sky images both in daylight and nighttime conditions. The paper explains the algorithm that is applied to derive cloud cover from the images of this camera. Moreover, a thorough validation-comparison effort is performed between cloud cover derived from these images, from images of other two (visible, that is, only during daylight hours) commercial sky cameras, and from the APCADA algorithm (based on cloud effect on downward longwave radiation measured with a pyrgeometer).**
**In some way this paper is a follow-on of a previous paper by the same research group (Aebi et al, 2017, AMT) where they presented an analysis of a long series of diurnal cloud cover obtained with a sky camera. The present paper, however, has several added values: the introduction of a new concept of an infrared sky camera (looking downwards to a convex mirror), the suggestion of the method for image processing, and the comparison with other estimations of cloud cover. Therefore, the paper is worth of being published in AMT. It seems to me that a few changes could be considered to make it more complete and to get higher impact in the scientific community, but even in the present version, the paper may be good enough to merit publication.**

**Suggested general change:**

**1. In order to make more significant the comparison among all estimations of cloud cover, authors could consider applying exactly the same horizon mask to all images. For example, they could use a mask for the part of the image that is below 70 deg. SZA (20 deg. over the "flat" horizon). In fact, even APCADA algorithm is unsensitive to clouds that are in the horizon, so using this mask for all images would make the comparison more homogeneous.**

Thanks for this comment. The authors are aware that the comparison of the cameras and APCADA are problematic when different horizon masks and different field of views are considered. However, the focus of the paper is mainly to present a new camera system (IRCCAM) and to show the possibilities of this new camera system. Thus we decided to not decrease the field of view of the IRCCAM due to the fact that one of the camera software is not able to detect clouds below 70°.

**Minor changes and technical details to be corrected:**

**2. The word "significant" is used several times in the manuscript. I have my doubts about this use, as no statistical tests are applied (at least, they are not mentioned). So I would suggest use "significant" with caution, as it has a meaning related to statistical tests. If possible, try to use another word. In page 11, line 25, it is said that a difference of 0.02 is statistically not significant, but with no reference to what statistical test is applied.**

We exchanged the word significant throughout the manuscript.

3.  In lines 13-24 tow different approaches for cloudiness estimation are summarized. But in my opinion they are not clearly differentiated. Calbó et al 2001 suggests a method based on pyranometer measurements (i.e., hemispheric measurement of solar irradiance), which is very different to Nephelo or Nubiscope, which are measuring in the infrared and in a narrow field of view. Please consider slightly modifying the writing of this paragraph.

We changed the writing of this paragraph (p. 2, l. 20ff.):

*.... Depending on the wavelength range, the presence of clouds alters the radiation measured at ground level (e.g. Calbo et al., 2001; Mateos Villàn et al., 2010). Calbo et al. (2001) and Dürr and Philipona (2004) both present different methodologies to determine cloud conditions from broadband radiometers. Other groups describe methodologies using instruments with a smaller spectral range. Such instruments are for example the infrared pyrometer CIR-7 (Nephelo) (Tapakis and Charalambides, 2013) or Nubiscope (Boers et al., 2010; Feister et al., 2010; Brede et al., 2017), ....*

4.  I wouldn't say that WSI is among the most common all-sky cameras (as it is indeed the TSI). The WSI is one of the pioneering cameras, and presents very interesting characteristics and developments, but, to my knowledge, is no usually commercialized and therefore, is not quite common.

We changed the sentences discussing the TSI and the WSI (p. 3, l. 8f.):

*The most common all-sky camera is the commercially available total sky imager (TSI) (Long et al., 2006). Another pioneering hemispherical cloud detection instrument is the whole sky imager (WSI) (Shields et al., 2013).*

5.  Eq (3). Could you explain why the zenith angle is divided by 65?

Equation 3 is a normalized function to fit the sky brightness temperature. Since we are taking the sky brightness temperature at 65° ($T_{65}$), we also divide by 65. Smith and Toumi, 2008 present the example to normalize at 90°, but in our case 90° is not representing the sky, but the mountains.

6.  It is interesting to note that different thresholds are used in the processing of the visible images. This could partly explain some of the difference found in this paper. In fact, selection of the threshold is critical to distinguish between a cloudy pixel, and a clear (but sometimes, containing aerosol) pixel. Some discussion on this matter may be found in Calbó et al 2017 (and other studies cited therein). [Calbó, J., C. N. Long, J. González, J. Augustine, and A. Mccomiskey, 2017: The thin border between cloud and aerosol: Sensitivity of several ground based observation techniques. Atmos. Res., 196, 248–260, doi:10.1016/j.atmosres.2017.06.010.]

We added the reference in the conclusion (p. 14, l. 10ff.):

*Differences in the cloud fraction estimates can be due to different thresholds for the camera systems (as discussed in Calbo et al. (2017)) as well as some other issues addressed throughout the current study.*

7. The authors recognize that IRCCAM fails at both extremes of the cloud cover distribution. This (unexpected) result should merit more attention, with a deeper discussion if possible.

We added a short discussion about a possible reason for this distribution (p. 10, l. 4ff.):

*It is noteworthy that the IRCCAM clearly underestimates the occurrence of 0 oktas in comparison to the cameras measuring in the visible spectrum (by up to 13 %). On the other hand, the relative frequency of the IRCCAM of 1 okta is clearly larger (by up to 10 %) compared to the visible cameras. This can be explained by higher brightness temperatures measured in the vicinity of the horizon above Davos. These higher measured brightness temperatures are falsely determined as cloudy pixels (up to 0.16 cloud fraction). Since these situations with larger brightness temperatures occur quite frequently, the IRCCAM algorithm detects more often cloud coverages of 1 okta instead of 0 okta.*

8. The first lines of section 3.2.3 (lines 21-29) do not address seasonal analyses, so I suggest moving them to another section.

We moved these lines to section 3.2.

9. Somewhere in the Results or Conclusion sections, I would appreciate a short discussion of the present results in comparison with performance of other IR whole sky cameras (if you can find any) or other sky cameras that take night images. If no previous work can be found with an estimation of the performance of such night images, this should be highlighted in the paper. Suggested references: [Shields, J. E., M. E. Karr, R. W. Johnson, and A. R. Burden, 2013: Day/night whole sky imagers for 24-h cloud and sky assessment: history and overview. Appl. Opt., 52, 1605–1616, doi:10.1364/AO.52.001605; Gacal, G.F.B. Antioquia, C., and N. Lagrosas, 2016: Ground-based detection of nighttime clouds using a digital camera. Appl. Opt., 55, 6040–6045, doi:10.1364/AO.55.006040.]

The authors could not find any comparisons to other IR whole sky camera systems.

10. Figure 4, caption and related text. There is some mistake in the definition of oktas from cloud fraction. According with the caption, 0 oktas is for cloud fraction between 0 and 0.05 (which looks correct to me) but 8 oktas is between 0.875 and 1.0, that is a much larger interval, which seems wrong (at least, it is not symmetrical). And, for example, 4 oktas should be 0.4375-0.5625 (that is, a bin centered in 0.5 with a width of 0.125). If you correct this, some differences among the methods may change.

Thanks for this comment. The description of the okta ranges in the caption of Figure 5 (before Figure 4) was indeed wrong and we corrected it. However, the analysis was done with the correct ranges.

11. Table 1 and table 2 could be put together in a single matrix-like table (like the authors do in Table 5). In each cell (only in one triangle of the matrix) both the median and the percentiles may be written (for example, as 0.01 [-0.24,0.21].

Since Table 1 and Table 2 are placed in the corresponding sections where they are discussed, we decided to not merge them.

---

## Referee Report (RR1)

**Summary**

The majority of major comments was addressed, but often not in the manuscript itself. Some new major comments emerged. There are a couple of problems with the references. Please provide a difference version for quicker review.

**Old major comments:**

7. Please provide a cost estimate of all used systems.

-> I could not find cost estimates in the text.

12. P. 3, l. 3: "low cost commercial cameras" "give no information during nighttime. There are several commercially available IR surveillance cameras out there. Are you sure that they are not used in meteorology as of today?

->  You might add here: "to the best of our knowledge" or use the phrase in minor comment 9. This statement might continue to be incorrect.

20. P6, l. 1: I was wondering about the mirror temperature and potential asymmetries. Could you briefly state if the one-sided heating of the sun leads to a temperature distribution on that mirror within your stated 1 K range? In Fig. 1, a wall is visible close to the IR-camera – is there a problem with radiated heat, e.g. during night-times? Could you briefly state something on the interplay between the ground temperature and that mirror? Do you expect aging effects on the mirror? **How bad is the soling?**

-> I could not find a statement regarding soling, which I think is a major issue for systems having a large upward-facing mirror with a holder placed above, inviting the local flying fauna to sit and digest there. Please indicate cleaning schedules. Was there any work towards automation?

30. Specify the total run time of each algorithm.

-> I could not find a statement on algorithmic run time in the text.

**New major comments:**

There a certain issues which are so far not addressed:

A. The theoretical problem to differentiate clouds and aerosols (Calbó et al., (2017), Glickman and Zenk (2000), etc.), having a major impact on the whole study.

B. The built system suffers from the self-occulsion effect of clouds as well as projection uncertainties, s. Fig. 1. You might briefly discuss the effects of these problems for your system.

[Figure]

**Figure 1: Left: Self-occlusion effect of clouds – the cloud coverage seems to increase towards the horizon. Right: Projection uncertainties of ground-based sky-imaging systems. You might cite reference 1 on that.**

References:

```
1.  @inproceedings{Kuhn2018VergleichundBewertungsolarerNowcasting-Systeme,
            author = {         Pascal Kuhn and Bijan Nouri, Stefan Wilbert and Laura Bianco
    and Loïc Vallance and Christoph Prahl and Lourdes Ramirez and Luis Zarzalejo and Thomas Schmidt
    and Zeyad Yasser and Laurent Vuilleumier and Detlev Heinemann and A. Kazantzidis and J. M.
    Wilczak and Philippe Blanc and Robert Pitz-Paal},
            editor = {conexio GmbH},
            title = {{Vergleich und Bewertung solarer Nowcasting-Systeme}},
            booktitle = {Tagungsunterlagen PV-Symposium 2018},
            date = {25.-27.04.2018},
            year ={2018},
            url={http://www.pv-symposium.de/programm/tagungsunterlagen.html}
}
```

C. Motivated by "B" – there are systems which do not suffer from these effects, such as https://doi.org/10.1016/j.solener.2017.05.074 / https://www.adv-sci-res.net/15/11/2018/ : Systems using downward-facing cameras are in that regard superior to sky-imaging systems. Please include these systems while introducing systems which are able to measure cloud coverages. Please also include approaches based on monitored PV plants, which might be nowadays a bit more commonly conducted than human observations. You might study publications of MeteoControl on that topic.

D. Include in the caption of Fig. 5 that for this comparison only timestamps are included during which all systems provided measurements.

**Minor comments:**

1. p.1, l. 24: "… e.g. mainly" - language

2. p.2, l.2, comma, "… and, due to visibility issues, …"

3. p.2. l. 3, "at high spatial" – language

4. p.2., l. 14, "large field of view of satellites" -> "limited resolution of satellites (currently 500~m max)"

5. p.2., 17, "Nowadays satellite data are validated" – I'm not sure if that was much different say 10 years ago. Please rephrase.

6. p.2., 19, semicolons

7. p.2., l. 34, "a laser pulse" – there is a certain tendency to use plural in such circumstances. There is also a comma missing, "…, send"

8. p.3, l. 6, " which often have a…"

9. p.3., l. 14, "often give limited information"

10. p.3., l.16, "cloud base height" – not shown in the paper, currently not accepted by everyone in the community, I suggest to leave it out. If kept, please provide an ISI-listed source on a corresponding multi-day validation.

11. p.4, section 2: Specify cleaning routines here

12. p.5., l.4. There is a change in style regarding "…'…" / "…,…" within the paper. Use one style.

13. p.11, l.25, 2x "to take"

14. p. 18, l. 21; "Bijan, Stefan, Nora …" -> these are not the family names

15. p. 18, l. 26; there also seems to be an issue with the names here

16. p. 18, l. 11; there is an issue with the doi

17. p. 19, l. 37; presumably an issue with capital letters in the title, same for Sields et al.

-> Check all your references

General: I'd like to use the opportunity to suggest to the journal placing the images where they belong, not at the end of the script. This might have had some reasons, but I don't think these reasons made it in this century.

---

## Author Response (AR2)

Dear editor

We thank you for your constructive comments on the manuscript "Cloud fraction determined by thermal infrared and visible all-sky cameras" by Aebi et al., submitted to Atmospheric Measurement Techniques (amt-2018-68).

The answers to your comments follow below (bold: editor comment, regular font: author's response, italic: changes in the manuscript).

Best regards

Christine Aebi

**Comments to the Author:**
**Dear Author,**

**in order to proceed with the review process of your manuscript, and to avoid patent infringement and concealment of anteriority, I would require the two following modifications:**

**1.) p.3 line 27 present Sky Insight as an industrial product and cite the patent. The patent can be cited in reference as follow: Bertin C., Cros S., Schmutz N., Liandrat O., Nicolas S., Lalire S., Detection unit and method for identifying and monitoring clouds in an observed area of the sky. Patent number FR3026496, WO2016046309, 2014-09-26.**

We changed the sentences about the instrument from Reuniwatt in the introduction and added a reference to the patent:

p. 3, l. 26ff.:

*The Sky Insight thermal infrared cloud imager is an industrial and patented (Bertin et al., 2015b) product from Reuniwatt. The Sky Insight cloud imager is sensitive in the 8 µm - 13 µm wavelength range and gives cloud information of the whole upper hemisphere. Their system is mainly used for cloud cover forecasts up to 30 minutes ahead, which is relevant for e.g. global horizontal irradiance forecasts or optical communication link availability (Bertin et al., 2015a; Liandrat et al., 2017).*

**2.) Section 2.1.1 p.7, an explicit mention in the text should express that the cloud detection algorithm is based on a similar approach as the one of Bertin et al., 2015:**

**Bertin C., Cros S., Saint-Antonin L., Schmutz N., "Prediction of optical communication link availability: real-time observation of cloud patterns using a ground-based thermal infrared camera", Proc. SPIE 9641, Optics in Atmospheric Propagation and Adaptive Systems XVIII, 96410A (8 October 2015); https://doi.org/10.1117/12.2194920**

We added the following sentence in our method section:

p. 6, l. 8f:

*A similar approach to detect cloud patterns is described in Bertin et al. (2015a) and Liandrat et al. (2017).*

To make it more clear that our method is not based on the method of Reuniwatt (as you suggested), we also added the following sentence with the two references:

p. 3, l. 34ff.

*After a developing and testing phase (Aebi et al., 2014; Gröbner et al., 2015), the IRCCAM is in continuous use at the Physikalisch-Meteorologisches Observatorium Davos/World Radiation Center (PMOD/WRC), Davos, Switzerland, since September 2015.*

**You may decide to submit a revised version of the manuscript after having included the above changes and taking into account the referee report.**

**Best regards, Manfred Wendisch**

[revised manuscript text omitted]